# Co-targeting JAK1/STAT6/GAS6/TAM signaling improves chemotherapy efficacy in Ewing sarcoma

Le Yu[1,2], Yu Deng [1,2], Xiaodong Wang[1,3], Charlene Santos[1], Ian J. Davis[1,4,5], H. Shelton Earp[1,6] & Pengda Liu [1,2] ✉

Ewing sarcoma is a pediatric bone and soft tissue tumor treated with chemotherapy, radiation, and surgery. Despite intensive multimodality therapy, ~50% patients eventually relapse and die of the disease due to chemoresistance. Here, using phospho-profiling, we find Ewing sarcoma cells treated with chemotherapeutic agents activate TAM (TYRO3, AXL, MERTK) kinases to augment Akt and ERK signaling facilitating chemoresistance. Mechanistically, chemotherapy-induced JAK1-SQ phosphorylation releases JAK1 pseudokinase domain inhibition allowing for JAK1 activation. This alternative JAK1 activation mechanism leads to STAT6 nuclear translocation triggering transcription and secretion of the TAM kinase ligand GAS6 with autocrine/paracrine consequences. Importantly, pharmacological inhibition of either JAK1 by filgotinib or TAM kinases by UNC2025 sensitizes Ewing sarcoma to chemotherapy in vitro and in vivo. Excitingly, the TAM kinase inhibitor MRX-2843 currently in human clinical trials to treat AML and advanced solid tumors, enhances chemotherapy efficacy to further suppress Ewing sarcoma tumor growth in vivo. Our findings reveal an Ewing sarcoma chemoresistance mechanism with an immediate translational value.

Ewing sarcoma, an aggressive malignancy of bones or soft tissues affecting primarily children and young adults, accounts for 15% of primary bone sarcomas[1]. Chemotherapy, radiation, and surgery constitute first-line therapy for Ewing sarcoma patients in the absence of targeted therapies. Despite intensive multimodality therapy, ~50% of patients eventually relapse and die of the disease due to developed therapy resistance[2]. Currently, there is no FDA-approved treatment regimen for relapsed Ewing sarcoma patients. Even for patients who respond to chemotherapy, adverse consequences of iterative high doses of chemotherapy significantly reduce patient quality of life. Thus, there is an urgent need for new therapeutic strategies for

relapsed patients, as well as to reduce severe adverse effects associated with current chemotherapy.

The critical driver of Ewing sarcoma results from a chromosomal translocation that brings together the *EWSR1* gene (or one of the other so-called FET genes) and one of several ETS transcription factors, most commonly *FLI1* (EWS::FLI1, in ~85% Ewing sarcoma patients)[3]. The chimeric EWS::FLI1 transcription factor maintains malignant transformation by associating with GGAA microsatellite regions to induce nucleosome depletion at these sites[4–6] and regulate transcription of multiple target genes including DAX1, GLI1, FOXOs, LOX, HD1, IGF1 and others[7]. In addition, EWS::FLI1 has also been implicated in R-loop

[1]Lineberger Comprehensive Cancer Center, The University of North Carolina at Chapel Hill, Chapel Hill, NC 27599, USA. [2]Department of Biochemistry and Biophysics, The University of North Carolina at Chapel Hill, Chapel Hill, NC 27599, USA. [3]Center for Integrative Chemical Biology and Drug Discovery, Division of Chemical Biology and Medicinal Chemistry, Eshelman School of Pharmacy, The University of North Carolina at Chapel Hill, Chapel Hill, NC 27599, USA. [4]Department of Genetics, The University of North Carolina at Chapel Hill, Chapel Hill, NC 27599, USA. [5]Department of Pediatrics, The University of North Carolina at Chapel Hill, Chapel Hill, NC 27599, USA. [6]Department of Medicine and Pharmacology, The University of North Carolina at Chapel Hill, Chapel Hill, NC 27599, USA. ✉e-mail: pengda_liu@med.unc.edu

formation and impairing BRCA1-mediated homologous recombination repair as another layer of mechanism facilitating tumorigenesis[8]. Recent studies indicated that higher levels of EWS::FLI1 proteins are associated with an immature and proliferative phenotype, whereas reduced EWS::FLI1 protein levels correlate with reduced proliferation and a more motile phenotype[9,10]. Previous efforts to therapeutically target the EWS::FLI1 oncoprotein focusing on targeting downstream genes have largely failed[11]. Ongoing efforts explore strategies to block EWS::FLI1 interaction with DNA[12] or aberrant chromatin accessibility[13]. Currently, TK216, that targets EWS::FLI1 and microtubules[14] and inhibitors for LSD1 (lysine-specific demethylase 1)[15], are under clinical evaluations, and their clinical performance remains unclear. As EWS::FLI1 occurs exclusively in tumor cells, the fusion protein offers a highly selective drug target. We recently reported that the SPOP E3 ligase and OTUD7A deubiquitinase govern EWS::FLI1 protein abundance through a ubiquitin-dependent protein stability control mechanism and demonstrated the activity of first-generation OTUD7A inhibitors in decreasing EWS::FLI1 protein levels and suppressing Ewing sarcoma growth in vitro and in animals[16]. Although promising therapeutic leads are beginning to emerge from these studies, the efficacy of these agents in clinical settings remains to be determined.

Single-agent therapies in cancer are typically less effective than combination treatments due to the reliance on a single mechanism for which resistance can emerge and the absence of therapeutic synergy. Ongoing clinical trials in Ewing sarcoma have recently focused on combination therapies to enhance active chemotherapy. Previous analysis of autocrine signaling circuits in Ewing sarcoma identified the activity of IGF1 (IGF2)/IGF1R signaling in both Ewing sarcoma cells and patient tissues. Although inhibiting IGF1R decreases tumor cell growth[17], anti-IGF1R antibodies, and small molecule inhibitors have not been successful in clinical trials[18] and lack synergy with vincristine[19]. This may be due to low patient response rates (−10–14%)[18], resistance to IGF1R blockade via reactivation of compensatory mechanisms[17,20], or other reasons. Therefore, other methods to enhance chemotherapeutic efficacy are needed.

In this work, we aim to identify chemotherapeutic resistance mechanisms and test therapeutic imports. We find chemotherapy induces JAK1 kinase activation to phosphorylate STAT6 in promoting GAS6 transcription and secretion, a ligand activating TAM kinases to potentiate Akt/ERK signaling. This signaling cascade provides protection for cells allowing for cell adaption to further develop chemoresistance. We find various kinases within this signaling can be targeted, including JAK1 and TAM by small molecule inhibitors to sensitize Ewing sarcoma to chemotherapy.

## Results

### Phospho-profiling identifies Akt and ERK activation in chemotherapy-treated Ewing sarcoma cells and tumors

Initial therapy for localized Ewing sarcoma involves five chemotherapeutic agents administered in an alternating pattern, including doxorubicin and etoposide. In the context of recurrence, irinotecan (which is metabolized to the topoisomerase I inhibitor, SN-38) and temozolomide (TMZ) have been shown to be temporarily effective for a large fraction of patients[21], and this combination has been used together with experimental agents[22,23]. Because of the relevance of these compounds for up-front and relapse therapy, we studied SN-38 (irinotecan for in vivo studies as a pro-drug), TMZ, etoposide, and doxorubicin in this investigation. We hypothesized that Ewing sarcoma cells acutely adapt to chemotherapy by altering specific signaling pathways prior to genomic changes. We performed phospho-profiling using control and SN-38-treated Ewing sarcoma MHH-ES-1 cell lysates (Fig. 1a) and identified significantly increased phosphorylation of a group of proteins (Fig. 1b, c). Proteins with increased phosphorylation upon SN-38 treatment were enriched in the Akt and ERK oncogenic signaling pathways (Fig. 1d). Notably, Akt-pT308 was increased in this profiling (Fig. 1b, c) while ERK-p42/p44 itself was not, which might be due to the

profiling utilized ERK-pT202/Y204/T185/Y187 antibody not suitable for directly measuring ERK-p42/p44 signals. We then examined whether activation of Akt/ERK signaling was seen with exposure to other chemotherapeutic agents by testing a broader range of Ewing sarcoma cells. To this end, we observed that SN-38 treatment increased Akt-pT308 and ERK-p42/p44 signals in a dose-dependent manner in A673 (Fig. 1e), MHH-ES-1 (Fig. 1f), TC-71 (Supplementary Fig. 1a) and TC-32 (Supplementary Fig. 1b) cells, and in a time-dependent manner in A673 (Supplementary Fig. 1c) and MHH-ES-1 (Supplementary Fig. 1d, 1e) cells. Similarly, treatment with TMZ (Fig. 1g, h and Supplementary Fig. 1f), etoposide (Fig. 1i, j and Supplementary Fig. 1g), and doxorubicin (Fig. 1k, l and Supplementary Fig. 1h) increased activation of both Akt and ERK signaling in multiple Ewing sarcoma cells in vitro. In addition to single chemotherapeutic agents, we also tested the clinical combination of SN-38 with TMZ in regulating Akt/ERK activation. Compared with single chemotherapeutic agents, the combination further increased ERK-p42/p44 with less effects on Akt-pT308 in A673 (Supplementary Fig. 1i), MHH-ES-1 (Supplementary Fig. 1j) or TC-71 cells (Supplementary Fig. 1k). Together, these data suggest that chemotherapeutic agents (both alkylator and topoisomerase inhibitors) can generally activate Akt/ERK signaling in Ewing sarcoma cells.

To investigate if chemotherapy also exerts a similar effect in Ewing sarcoma tumors, we xenografted A673 cells into nude mice and treated established Ewing sarcoma tumors with TMZ. TMZ treatment reduced tumor growth in nude mice (Fig. 1m–o) with minimal effects on animal weights and health (Supplementary Fig. 2a). Importantly, xenografted tumors received TMZ showed increased Akt-pT308 and ERK-p42/p44 levels compared with vehicle control-treated tumors (Fig. 1p and Supplementary Fig. 2b). In addition, increased activation of Akt and ERK was also observed in xenografted A673 tumors treated with irinotecan (the pro-drug of SN-38) (Supplementary Fig. 2c–e). These data suggest that chemotherapeutic agents may also activate Akt/ERK signaling in Ewing sarcoma tumors in vivo.

### Inhibiting TAM kinases alleviates chemotherapy-induced activation of Akt and ERK in Ewing sarcoma

Although multiple studies have suggested the roles of Akt and ERK in promoting drug resistance in a variety of cancer types[24–26] through distinct molecular mechanisms, the effects of these kinases on chemotherapy response in Ewing sarcoma have not been described. We treated MHH-ES-1 cells with either an Akt inhibitor (MK2206) or an ERK inhibitor (PD0325901). Both inhibitors sensitized MHH-ES-1 cells to etoposide (Fig. 2a, b), suggesting that inhibition of these signaling pathways may enhance the effect of chemotherapy to treat Ewing sarcoma. However, neither Akt nor ERK inhibitors have shown satisfactory clinical effects in cancer therapy due to toxicity resulting from inhibition of their essential physiological roles[26]. Using a panel of kinase inhibitors with diverse activities, we tested for upstream kinase(s) that activate Akt/ERK selectively following chemotherapeutic exposure that may serve as targets in Ewing sarcoma. Two compounds, UNC2025[27] and UNC5293[28], suppressed SN-38-induced Akt/ERK activation in A673 cells (Fig. 2c). Both compounds were developed at UNC to inhibit TAM (TYRO3, AXL, and MERTK) kinases[27]. These data implicate that TAM kinases may govern chemotherapy-induced Akt/ERK activation in Ewing sarcoma.

Because these two compounds both target MERTK, but can engage other TAM and kinase targets[27], to test specificity, next, we genetically depleted each of TYRO3, AXL and MERTK individually to examine their contributions to chemotherapy-induced activation of Akt/ERK signaling. Depletion of MERTK or TYRO3, but not AXL, in MHH-ES-1 cells significantly reduced Akt-pT308 and ERK-p42/p44 signals induced by SN-38 (Fig. 2d–f), doxorubicin (Supplementary Fig. 3a–c), etoposide (Supplementary Fig. 3d–f) or TMZ (Supplementary Fig. 3g–i). This was further confirmed in A673 cells depleted of each TAM kinase member challenged with SN-38 (Supplementary Fig. 3j–l). Importantly, depletion of neither AXL nor TYRO3 significantly reduced basal activities of Akt

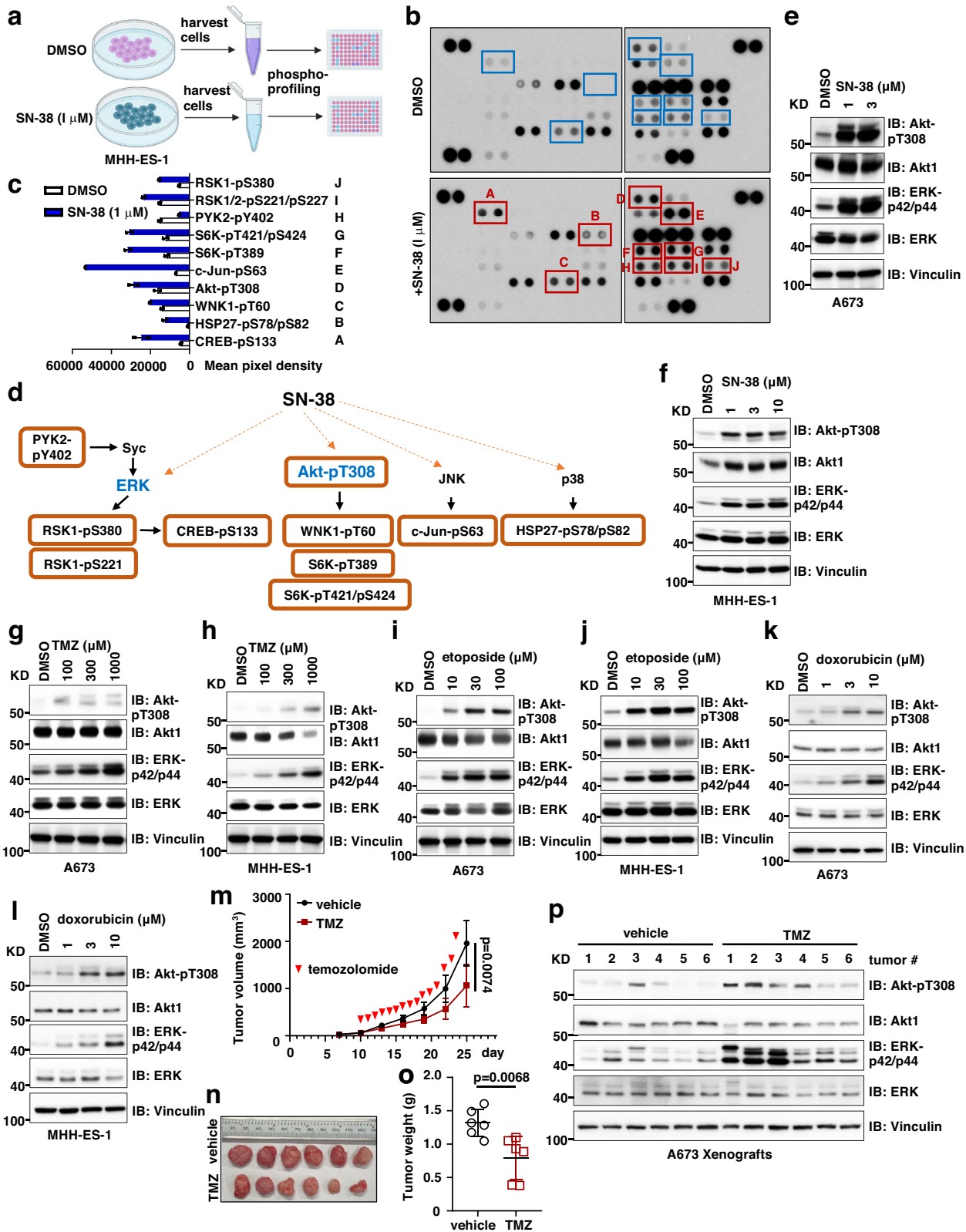

and ERK under non-treatment conditions, while MERTK depletion mildly reduced ERK with neglectable effects on Akt activation (Fig. 2d–f). The unique MERTK/TYRO3 mediated activation of Akt/ERK signaling upon chemotherapeutic treatment suggests MERTK/TYRO3 inhibition as a therapeutic option compared with canonical Akt or ERK inhibitors (that suppresses Akt or ERK activation under non-treatment conditions). These data suggest that MERTK and TYRO3, but not AXL

govern chemotherapy-induced activation of Akt and ERK signaling and contribute to chemoresistance. Consistent with signaling data, depletion of either MERTK, AXL or TYRO3 sensitized MHH-ES-1 cells to SN-38 (Fig. 2g–i) or etoposide (Fig. 2j–l). Interestingly, depletion of AXL reduced MHH-ES-1 2D colony formation ability in vitro (Supplementary Fig. 3m–o), depletion of MERTK slightly reduced MHH-ES-1 colony formation (Supplementary Fig. 3p–r), while depletion of TYRO3 did not

**Fig. 1 | Chemotherapeutic treatments induce activation of Akt and ERK in Ewing sarcoma. a** A cartoon illustration of the experimental procedure for the phospho-profiling experiment generated by Biorender. Please refer to the Method section for details. **b** The signals from the phospho-profiling assay. Increased phosphorylation signals are boxed and labeled with letters. **c, d** Quantifications of the phospho-profiling signals in (**b**) and representative signals for each letter from (**b**) and grouped in (**d**). Error bars were calculated as mean ± SD, $n = 2$ (technical duplicates). **e, f** Immunoblot (IB) analysis of whole cell lysates (WCL) from A673 cells (**e**) or MHH-ES-1 cells (**f**) treated with indicated doses of SN-38 for 24 h. **g, h** IB analysis of WCL from A673 cells (**g**) or MHH-ES-1 cells (**h**) treated with indicated doses of TMZ (temozolomide) for 24 h. **i, j** IB analysis of WCL from A673 cells (**i**) or MHH-ES-1 cells (**j**) treated with indicated doses of etoposide for 24 h. **k, l** IB analysis of WCL from A673 cells (**k**) or MHH-ES-1 cells (**l**) treated with indicated doses of doxorubicin for 24 h. **m** Tumor volume measurements at indicated days post-injection with temozolomide injected at the indicated days. Temozolomide (50 mg/kg) were injected via IP. Error bars were calculated as mean ± SD, $n = 6$ tumors. $p$ values are calculated as indicated (two-way ANOVA followed by Tukey multiple comparison test). **n** Isolated tumors from (**m**) and weighed in (**o**). Error bars were calculated as mean ± SD, $n = 6$ tumors. $p$ values are calculated as indicated (two-tailed student's $t$-test). **p** IB analysis of WCL derived from dissected tumors with or without Temozolomide treatment. The number indicates each tumor obtained from animals.

affect MHH-ES-1 cell proliferation at all (Supplementary Fig. 3s–u). These data suggest that AXL may control cell proliferation independent of Akt/ERK signaling.

We also examined the roles of IGF1R in chemotherapy-induced activation of Akt/ERK signaling. Depletion of IGF1R reduced Akt-pT308 and ERK-p42/p44 signals in MHH-ES-1 cells (Supplementary Fig. 4a), accompanied by reduced 2D colony formation ability in vitro (Supplementary Fig. 4b, c). However, IGF1R depletion failed to block SN-38-induced ERK/Akt activation (Supplementary Fig. 4d), which is consistent with the observation that IGF1R inhibition didn't exert a synergy with chemotherapy in clinical trials[19]. However, additional mechanisms beyond Akt/ERK regulation may also account for IGF1R targeting in modulating Ewing sarcoma chemotherapeutic responses. In addition, this may also suggest that chemoresistance in Ewing sarcoma can be separated from cell growth status.

## UNC2025 or UNC5293 synergizes with chemotherapeutics to reduce Ewing sarcoma growth

Given that chemotherapy induces Akt/ERK activation (Fig. 1b–d), which could be suppressed by TAM kinases inhibitions via UNC2025 or UNC5293 (Fig. 2c), and genetic depletion of either MERTK or TYRO3 sensitized Ewing sarcoma cells to SN-38 or etoposide (Fig. 2g–k), we examined whether UNC2025 or UNC5293 synergizes with chemotherapy to reduce Ewing sarcoma growth. We first tested 8 combinations (including two TAM kinase inhibitors UNC2025 or UNC5293, and four chemotherapeutic agents SN-38, etoposide, doxorubicin, and TMZ) in two Ewing sarcoma MHH-ES-1 and A673 cells (Fig. 3a, b and Supplementary Fig. 5a–o). We found the combination of UNC2025 + SN-38 showed the best synergy in reducing the proliferation of both cell lines in vitro. We next expanded our evaluation of this combination to a larger panel of Ewing sarcoma cells including TC-71, CHLA-10, TC-32, SK-N-MC, EWS894, and CHLA-32 (Fig. 3c–h). Using the Chou–Talalay method based on the median-effect equation for combination index (CI) determination[29] to evaluate the synergistic effects from combination treatments, we observed a similar synergistic effect in each of these Ewing sarcoma cells. Based on these promising in vitro data, we examined the efficacy of this combination in vivo. Mice bearing MHH-ES-1 xenografts were treated with vehicle, UNC2025 (10 mg/kg IP), irinotecan (10 mg/kg, the pro-drug for SN-38, IP), and the combination. The relative low-dose treatment of UNC2025 alone (10 mg/kg IP here vs. 50 or 75 mg/kg oral commonly used[30]) slowed down MHH-ES-1 tumor growth in mice, and significantly enhanced the therapeutic effect of irinotecan as compared with single-agent treatment (Fig. 3i–k), supporting the therapeutic potential of the combination. IHC analysis on collected tumors suggested UNC2025 treatment efficiently reduced ERK-p42/p44 signals induced by irinotecan (Supplementary Fig. 5p).

## Chemotherapy induces GAS6 expression and secretion to facilitate TAM kinase activation

We then explored the mechanism(s) for TAM kinase inhibition-induced chemo-sensitization in Ewing sarcoma. TAM kinase activation is maximized by a complex ligand PtdSer (phosphatidylserine) externalized on

apoptotic materials and a bridging soluble protein factor GAS6 (for MERTK and AXL activation) or PROS (protein S, for MERTK and TYRO3 activation)[31,32] (Fig. 4a). PtdSer is a product of apoptosis by exposing inner layers of plasma membrane lipid bilayers. Similar to other cancers, chemotherapy yielded a source of PtdSer as treatment with SN-38 (Fig. 4b and Supplementary Fig. 6a, b), etoposide (Fig. 4c and Supplementary Fig. 6c), or doxorubicin (Supplementary Fig. 6d, e) led to Ewing sarcoma cell apoptosis as demonstrated by increased cleaved-PARP and cleaved-caspase 3, which could be prevented by Z-VAD-FMK (a pan caspase inhibitor, Fig. 4d). Inhibition of either Akt (by MK2206), ERK (by PD0325901) or TAM kinases (by UNC2025) further enhanced SN-38-induced apoptosis (Supplementary Fig. 6f). FACS analysis by Annexin V to directly measure PtdSer levels confirmed that both SN-38 and etoposide treatments significantly increased PtdSer levels (Fig. 4e). Interestingly, we found that the bridging ligand, GAS6, exhibited increased transcription induced by chemotherapeutic agents (SN-38, etoposide and doxorubicin) in both MHH-ES-1 and TC-32 cells (Fig. 4f, g); however, PROS transcription was not as much regulated as GAS6 by chemotherapeutics tested (Fig. 4h and Supplementary Fig. 6g). These data suggest that in Ewing sarcoma, chemotherapy provides both PtdSer and GAS6 (but not PROS) to activate TAM kinases.

To further examine if chemo-induced GAS6 expression contributes to TAM kinase activation in Ewing sarcoma, we next monitored GAS6 protein expression upon chemo-treatment. To our surprise, treatment with SN-38 (Fig. 4i), etoposide (Fig. 4j) or doxorubicin (Supplementary Fig. 6h) significantly reduced cellular GAS6 protein abundance (Fig. 4i, j), albeit at the same time GAS6 transcription was induced (Fig. 4f). Since GAS6 is a secreted protein, we measured cellular and media GAS6 protein levels by evaluating cell lysates or TCA-precipitated proteins from culture media. We found that while SN-38 (Fig. 4k and Supplementary Fig. 6i), etoposide (Fig. 4l), or doxorubicin (Supplementary Fig. 6j) reduced cellular GAS6 protein levels, treatments increased levels of secreted GAS6, providing autocrine or paracrine ligands to activate TAM kinases. To determine if GAS6 contributes to chemo-induced TAM kinase activation, we reduced endogenous GAS6 production using multiple independent GAS6 sgRNAs (Fig. 4m). GAS6 depletion alleviated activation of Akt and ERK induced by SN-38 (Fig. 4n) or etoposide (Fig. 4o). Consequently, depleting GAS6 sensitized MHH-ES-1 cells to SN-38 (Fig. 4p) or etoposide (Fig. 4q). Together, these data suggest GAS6 is necessary for chemotherapy-induced activation of Akt/ERK in Ewing sarcoma cells. To further examine the sufficiency of GAS6 in activating TAM kinases and subsequent chemoresistance, we prepared GAS6-conditioned media[33] (Fig. 4r). MHH-ES-1 cells treated with GAS6-conditioned media demonstrated an resistance to SN-38 (Fig. 4s) or etoposide (Fig. 4t), as GAS6-conditioned media (containing secreted GAS6 proteins, Supplementary Fig. 6k) activated TAM/Akt/ERK signaling in the absence of chemotherapy (Fig. 4u).

## STAT6 governs GAS6 transcription in response to chemotherapy in Ewing sarcoma

We next examined the transcription factor(s) governing GAS6 transcription upon chemotherapy treatment in Ewing sarcoma (Fig. 5a).

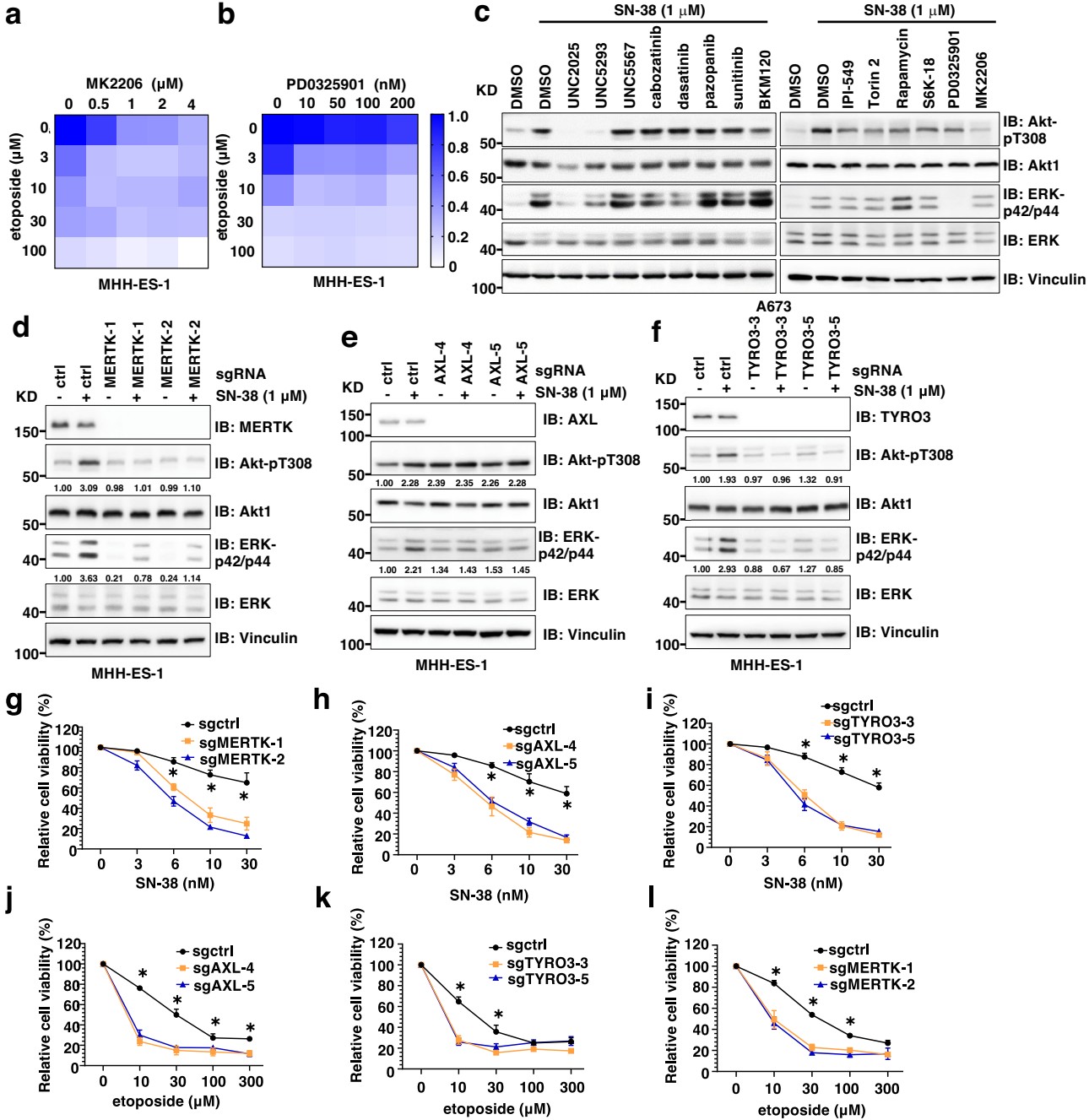

**Fig. 2 | MERTK and AXL mediate chemotherapy-induced activation of Akt and ERK in Ewing sarcoma. a**, **b** Representative heatmaps for cell viability in MHH-ES-1 cells treated with indicated doses of two compounds (as indicated in X and Y axes, respectively) for 2 days. The color scale bar indicates % of survived cells, where 1.0 represents 100%. **c** IB analysis of WCL derived from A673 cells treated with indicated compounds for 2 h followed by the addition of 1 μM SN-38 overnight before cell collection. **d**–**f** IB analysis of WCL derived from MHH-ES-1 cells depleted of MERTK (**d**), AXL (**e**), or TYRO3 (**f**). Where indicated, MHH-ES-1 cells were infected with indicated sgRNA viruses and selected in 1 μg/mL puromycin to eliminate non-infected cells for 72 h before cell collection. **g**–**l** Cell viability assays using indicated MHH-ES-1 cells treated with indicated doses of chemotherapeutic agents for 48 h. Error bars were calculated as mean ± SD, $n = 3$ (experimental triplicates). *$p < 0.05$

represents differences between the experimental groups compared to the control group (one-way ANOVA test). *$p$ value of each point (**g**: sgMERTK/sgctrl: 0.0012&0.0004 (6 nM); 0.0012 & <0.0001 (10 nM); 0.0038&0.0007 (30 nM); **h**: sgAXL/sgctrl: 0.002&0.0002 (6 nM), 0.0007&0.0013 (10 nM), 0.0004&0.0005 (30 nM); **i**: sgTYRO3/sgctrl: 0.0004&0.0003 (6 nM), <0.0001& < 0.0001 (10 nM), <0.0001& < 0.0001 (30 nM); **j**: sgAXL/sgctrl: <0.0001&0.0001 (10 nM), 0.001&0.0006 (30 nM), 0.0127&0.0147 (100 nM), 0.001&0.0008 (300 nM); **k**: sgTYRO3/sgctrl: 0.0004&0.0003 (10 nM), 0.0066&0.0236 (30 nM); **l**: sgMERTK/ sgctrl: 0.0026&0.0005 (10 nM), 0.0001 & <0.0001 (30 nM), 0.0009 & 0.0006 (100 nM). WB data presented in this figure are representative data from experimental duplicates.

Previous studies indicate multiple transcription factors including estrogen[34], NF-Y, ABFB, and BBF[35] may control GAS6 transcription in human or mouse cells. Since no reports indicating a role of estrogen[34] in Ewing sarcoma, we excluded it as a candidate to regulate GAS6 here. Interestingly, a recent study reported that in macrophages, activation

of IL-4/STAT6 signaling promoted GAS6 production to facilitate efferocytosis[36]. We evaluated whether STAT6 or NF-Y contributes to chemotherapy-induced GAS6 secretion to activate TAM/Akt/ERK signaling. Depletion of STAT6 (Supplementary Fig. 7a) largely abolished activation of Akt and ERK following either SN-38 (Fig. 5b) or etoposide

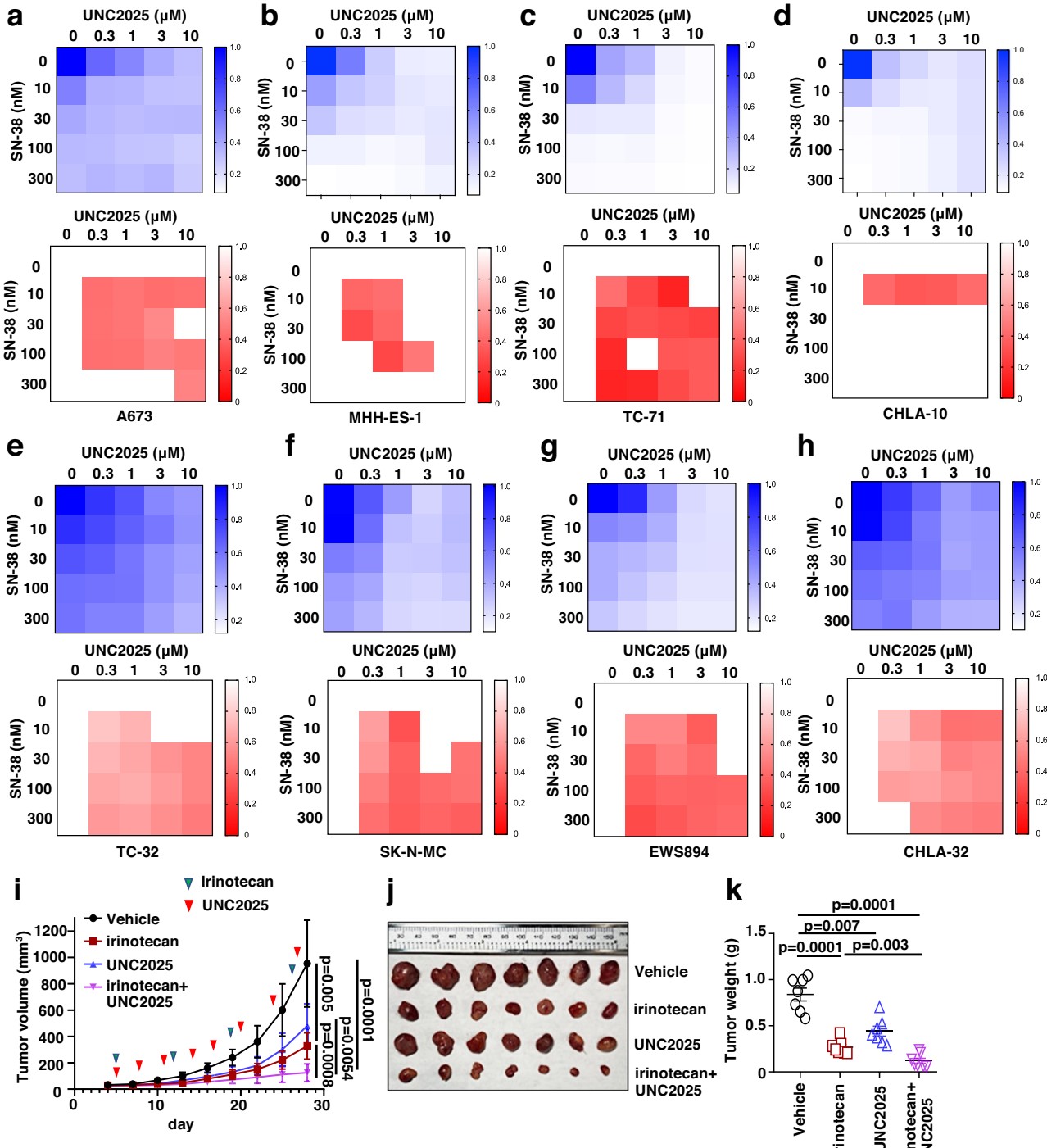

**Fig. 3 | UNC2025 sensitizes Ewing sarcoma cells to SN-38 in vitro and irinotecan in vivo. a–h** Representative heatmaps for cell viability (upper panels) and combination index (CI, lower panels) in MHH-ES-1 cells treated with indicated doses of UNC2025 (X-axis) with SN-38 (Y-axis) for 2 days. Please refer to the method section for more details. The heatmaps presented are representative data from biological triplicates. Cell viability was measured by Cell Titer-Glo assays and normalized to DMSO treatment. The color scale bar indicates % of survived cells, where 1.0 represents 100%. **i** Tumor volume measurements at indicated days post-injection with indicated compounds injected at the indicated days. Both irinotecan (10 mg/kg) and UNC2025 (10 mg/kg) were injected via IP. Error bars were calculated as mean ± SD, *n* = 7 tumors. *p* values are calculated as indicated (two-way ANOVA followed by Tukey multiple comparison test). **j** Isolated tumors from (**i**) and weighed in (**k**). Error bars were calculated as mean ± SD, *n* = 7 tumors. *p* values are calculated as indicated (two-tailed student's *t*-test).

(Fig. 5c) exposure. STAT6 depletion similarly blocked SN-38-induced Akt-pT308 and ERK-p42/p44 signals at higher SN-38 concentrations (10–300 μM) (Supplementary Fig. 7b). In contrast, depletion of NF-Y failed to do so (Supplementary Fig. 7c). Furthermore, in MHH-ES-1 cells depleted of endogenous STAT6, SN-38 treatment no longer induced GAS6 transcription (Fig. 5d) nor GAS6 secretion (Fig. 5e). In addition,

consensus binding sites for STAT6 in 5' regulatory region of the human *GAS6* gene was analyzed with Eukaryotic promoter database (SIB, Switzerland). Four potential STAT6 binding motifs were predicted within −2kb region with a *p* value smaller than 0.001, including SB1(−353), SB2(−487), SB3(−1440), and SB4(−1706) (Fig. 5f). By cut&run assays using STAT6 antibodies in control or SN-38 treated

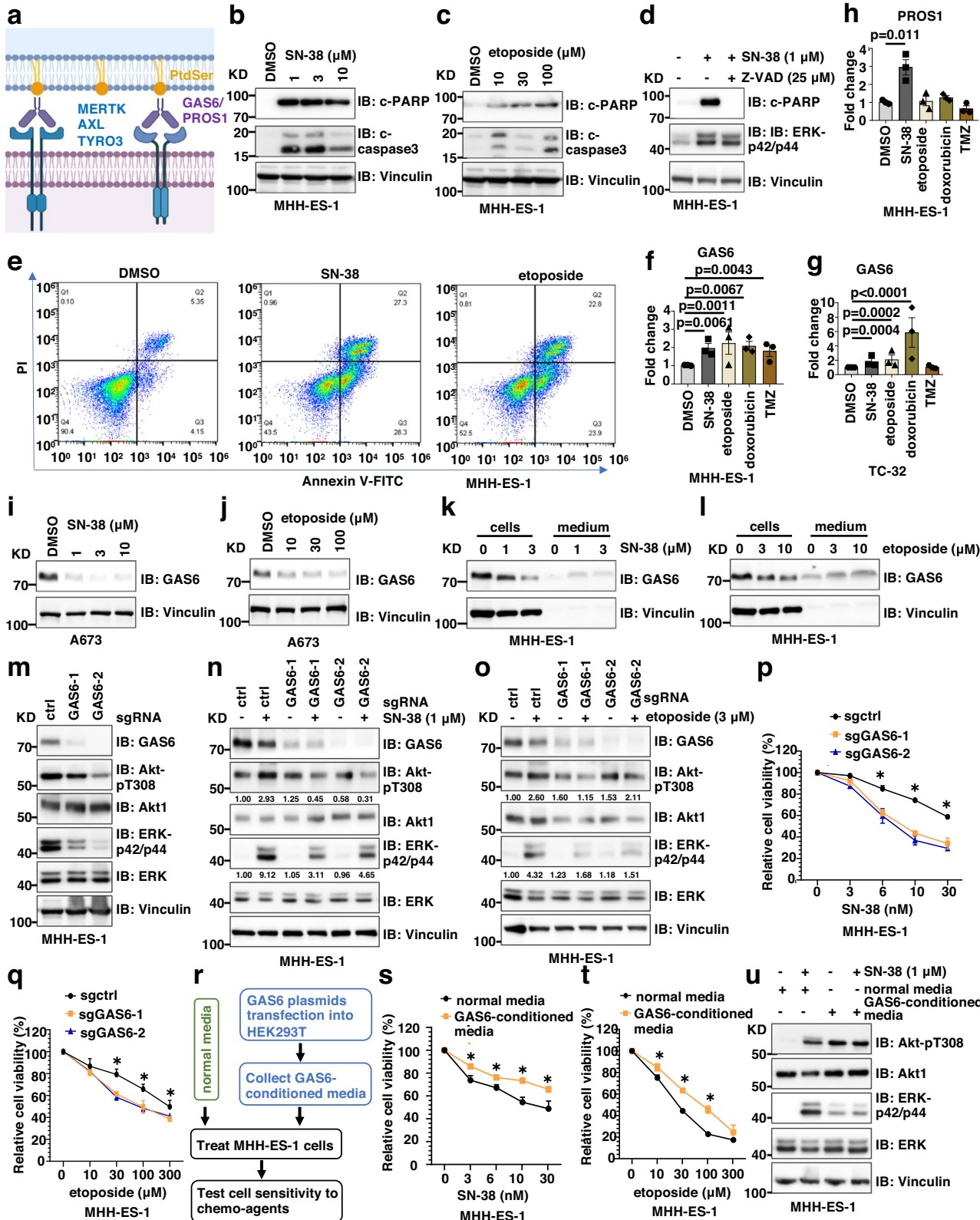

MHH-ES-1 cells, we found (1) STAT6 bound SB1 and SB3; (2) SN-38 significantly enhanced GAS6 binding to SB1 and SB3 (Fig. 5f). These data suggest that STAT6 may recognize SB1 and SB3 motifs in GAS6 promoter to facilitate SN-38-induced GAS6 transcription. Furthermore, like GAS6 depletion (Fig. 4p, q), depletion of STAT6 also sensitized MHH-ES-1 cells to SN-38 treatment (Fig. 5g). Importantly, STAT6 depletion-caused sensitization of MHH-ES-1 cells to etoposide could be partially reversed by treatment with GAS6-conditioned media (Fig. 5h).

Taken together, these data support the notion that STAT6 contributes to Ewing sarcoma chemotherapy resistance, in part, by regulating GAS6 transcription (Fig. 5a).

**Chemotherapy induces STAT6-pY641 and subsequent STAT6 nuclear translocation in a JAK1-dependent manner**

Since our data reveal that STAT6 is a regulator of GAS6 transcription in response to chemotherapy, we next examined how chemotherapy

**Fig. 4 | Chemotherapy induces GAS6 transcription and secretion to activate TAM kinases. a** A cartoon illustration of TAM kinase activating mechanisms prepared by Biorender. **b**–**d** IB analysis of WCL from MHH-ES-1 cells treated with indicated doses of SN-38 (**b**), etoposide (**c**), or SN-38 ± Z-VAD (**d**) for 24 h. **e** Representative FACS analyses of MHH-ES-1 cells treated with indicated compounds for 24 h. **f**–**h** RT-PCR analyses of changes of GAS6 (**f, g**) or PROS (**h**) in MHH-ES-1 cells treated with indicated chemotherapeutic agents for 24 h. SN-38, 1 μM; etoposide, 10 μM; doxorubicin, 1 μM; TMZ, 100 μM. Error bars were calculated as mean ± SD, $n = 3$ (experimental triplicates). $p$ values are labeled and represent differences between experimental groups compared to the control group (one-way ANOVA test). **i, j** IB analysis of WCL from MHH-ES-1 cells treated with indicated doses of SN-38 (**I**) or etoposide (**j**) for 24 h. **k, l** IB analysis of both WCL and TCA-precipitated culture media from MHH-ES-1 cells treated with indicated doses of SN-38 (**k**) or etoposide (**l**) for 24 h. **m** IB analysis of WCL derived from MHH-ES-1 cells depleted of GAS6. **n, o** IB analysis of WCL from indicated MHH-ES-1 cells with or without indicated doses of SN-38 (**n**) or etoposide (**o**) for 24 h. **p, q** Cell viability assays using control or GAS6-depleted MHH-ES-1 cells treated with

indicated doses of SN-38 (**p**) or etoposide (**q**) for 48 h. Error bars were calculated as mean ± SD, $n = 3$ (experimental triplicates). *$p < 0.05$ represents differences between the experimental groups compared to the control group (one-way ANOVA test). *$p$ value of each point of **p**: sgGAS6/sgctrl: 0.0007&0.0044 (6 nM); <0.0001&0.0002 (10 nM), 0.002<0.0001 (30 nM); **q**: sgGAS6/sgctrl: 0.0046&0.0037 (30 μM), 0.0174&0.004 (100 μM), 0.039&0.0798 (300 μM); **s**: 0.0098, 0.0105, 0.0019, 0.0141; **t**: 0.0237, <0.0001, 0.0007. **r** A cartoon illustration of the experimental procedure for the test of the contribution of secreted GAS6 in regulating MHH-ES-1 chemoresistance in vitro. **s, t** Cell viability assays using MHH-ES-1 cells pre-treated with normal media or GAS6-conditioned media for 24 h followed by treatment by indicated doses of SN-38 (**s**) or etoposide (**t**) for 72 h. Error bars were calculated as mean ± SD, $n = 3$ (experimental triplicates). *$p < 0.05$ represents differences between the experimental groups compared to the control group (one-way ANOVA test). **u** IB analysis of WCL from MHH-ES-1 cells pre-treated with normal media or GAS6-conditioned media for 24 h followed by treatment by 1 μM SN-38 for 24 h. WB data presented in this figure are representative data from experimental duplicates.

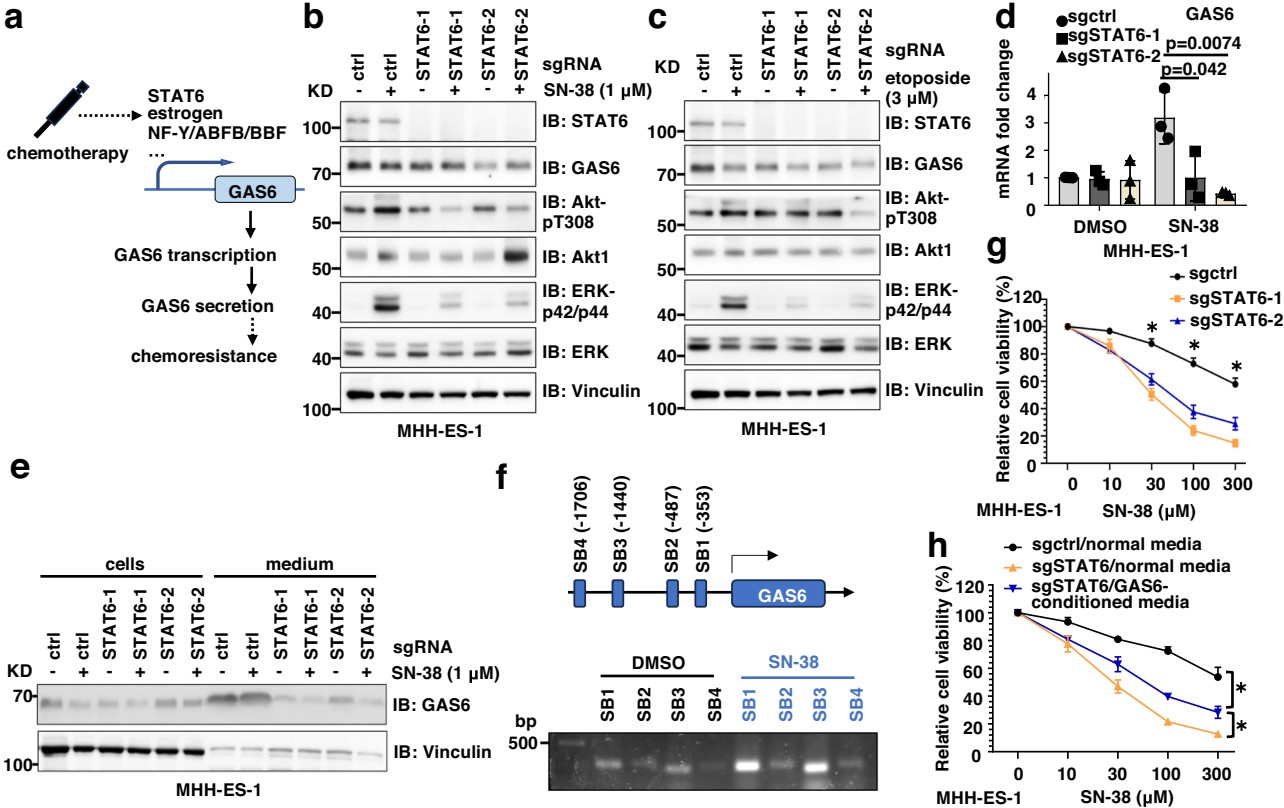

**Fig. 5 | STAT6 governs chemotherapy-induced GAS6 transcription in Ewing sarcoma. a** A cartoon illustration to reveal the involvement of transcription factors in mediating chemotherapy-induced GAS6 transcription in Ewing sarcoma. **b, c** IB analysis of WCL from control or STAT6-depleted MHH-ES-1 cells treated with or without indicated doses of SN-38 (**b**) or etoposide (**c**) for 24 h. **d** RT-PCR analyses of mRNA expression changes of GAS6 in control or STAT6-depleted MHH-ES-1 cells treated with SN-38 (1 μM) for 24 h. $p$ values are labeled and represent differences between experimental groups compared to the control group (one-way ANOVA test). **e** IB analysis of both WCL and BCA-precipitated culture media from control or STAT6-depleted MHH-ES-1 cells treated with SN-38 (1 μM) for 24 h. Error bars were calculated as mean ± SD, $n = 3$ (experimental triplicates). *$p < 0.05$ represents differences between the experimental groups compared to the control group (one-way ANOVA test). **f** upper: an illustration of predicted STAT6 binding sites on human GAS6 promoter. Lower: representative PCR analysis for STAT6 cut&run from MHH-ES-1 cells treated with control or 1 μM SN-38 for 24 h. **g** Cell viability

assays using control or STAT6-depleted MHH-ES-1 cells treated with the indicated dose of SN-38 (1 μM) for 48 h. Error bars were calculated as mean ± SD, $n = 3$ (experimental triplicates). *$p < 0.05$ represents differences between the experimental groups compared to the control group (one-way ANOVA test). *$p$ value of each point of **g**: sgSTAT6/sgctrl: 0.0003&0.0008 (30 μM), 0.0001&0.0007 (100 μM), 0.0001&0.0015 (300 μM). **h** Cell viability assays using control or STAT6-depleted MHH-ES-1 cells treated with the indicated dose of SN-38 (1 μM) for 48 h. Where indicated, STAT6-depleted MHH-ES-1 cells were pre-treated with GAS6-conditioned media for 24 h prior to SN-38 treatment. Error bars were calculated as mean ± SD, $n = 3$ (experimental triplicates). *$p < 0.05$ represents differences between the experimental groups compared to the control group (one-way ANOVA test). *$p$ value of each point of **h**: sgSTAT6-GAS6-conditioned media/sgctrl-normal media: <0.0001 (300 μM), sgSTAT6-GAS6-conditioned media/ sgSTAT6-normal media: 0.0118 (300 μM). WB data presented in this figure are representative data from biological duplicates.

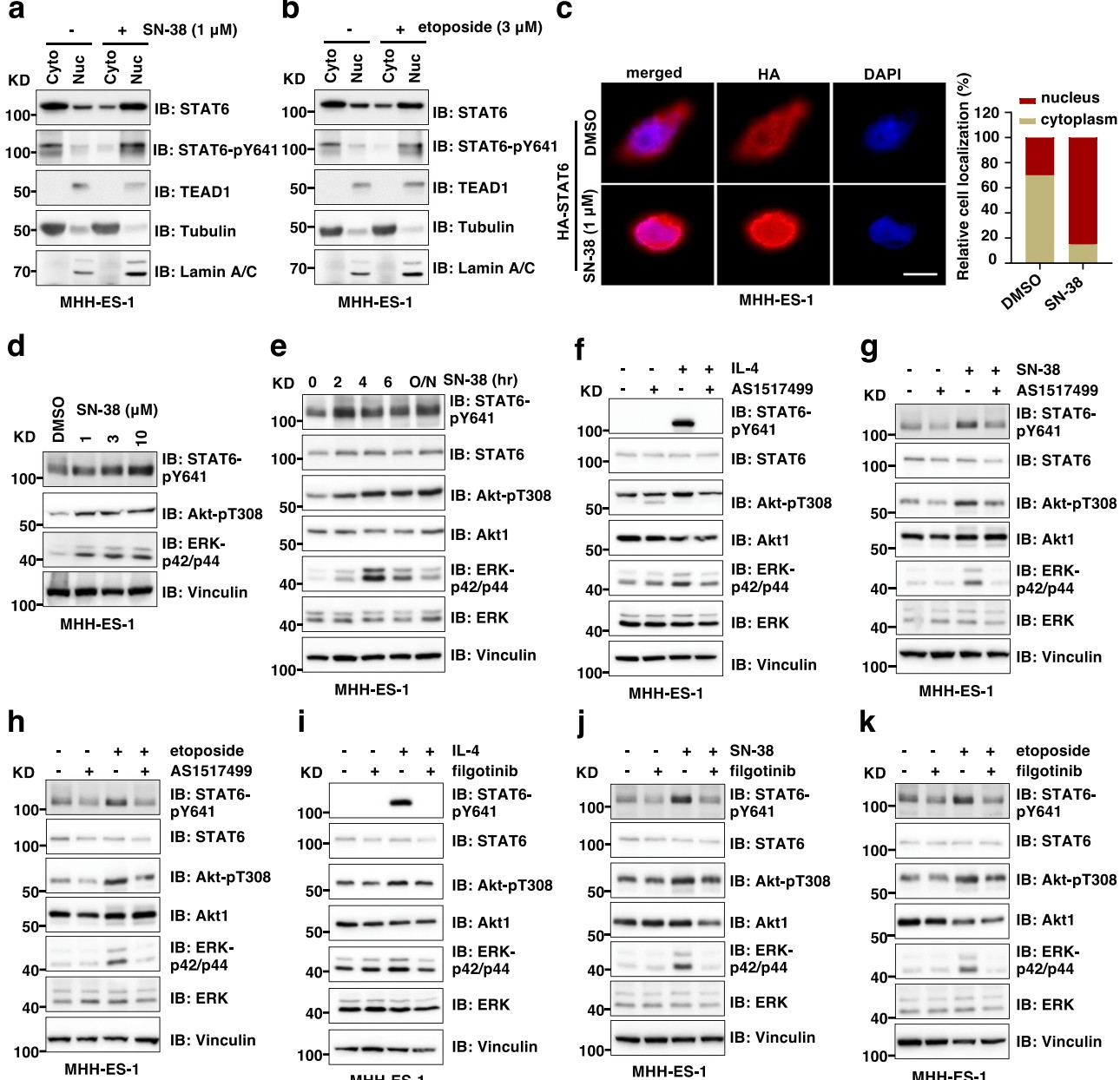

**Fig. 6 | Chemotherapy induces JAK1-dependent STAT6-Y641 phosphorylation and nuclear translocation. a, b** IB analysis of cytoplasmic and nuclear fractions from MHH-ES-1 cells treated with SN-38 (**a**) or etoposide (**b**) for 24 h. Cyto cytoplasm, Nuc nucleus. For quantification, STAT6 localization in cytoplasm or nuclei is determined by relative fluorescent intensity and at least 100 cells were used in each quantification. **c** Representative IF images from biological triplicates using MHH-ES-1 cells stably expressing EV or HA-STAT6 treated with or without SN-38 (1 μM) for 24 h. The scale bar represents 20 μm. The scale bar represents 20 μm. **d, e** IB analysis

of WCL from MHH-ES-1 cells treated with indicated doses of SN-38 (**d**) or 1 μM SN-38 for indicated time periods (**e**). **f–h** IB analysis of WCL from MHH-ES-1 cells treated with indicated compounds for 24 h. IL-4, 1 ng/mL; AS1517499, 5 μM; SN-38, 1 μM; etoposide, 10 μM. **i–k** IB analysis of WCL from MHH-ES-1 cells treated with indicated compounds for 24 h. IL-4, 1 ng/mL; filgotinib, 5 μM; SN-38, 1 μM; etoposide, 10 μM. WB data presented in this figure are representative data from experimental duplicates.

modulates STAT6 transcriptional activity. To this end, using cell fractionation assays, we found either SN-38 (Fig. 6a) or etoposide (Fig. 6b) significantly induced STAT6 nuclear translocation in MHH-ES-1 cells, which was accompanied by enriched STAT6-pY641 in nuclei (Fig. 6a, b). Independent IF experiments using MHH-ES-1 cells stably expressing HA-STAT6 (Supplementary Fig. 8a) also confirmed that SN-38 treatment induced STAT6 nuclear translocation (Fig. 6c). STAT6 is primarily activated by Th2 cytokines including IL-4 or IL-13 through phosphorylation by JAK1 at the Y641 residue. STAT6-pY641 is critical for STAT6 nuclear translocation and transcriptional activation[37,38], thus it is a hallmark of STAT6 activation. In MHH-ES-1

cells, we found SN-38 induced STAT6-pY641 accompanied by increased Akt-pT308 and ERK-p42/p44 signals in both SN-38 treatment dose- (Fig. 6d) and time- (Fig. 6e) dependent manners. Etoposide treatment led to similarly increased STAT6-pY641/Akt-pT308/ERK-p42/p44 signals in MHH-ES-1 cells (Supplementary Fig. 8b). A STAT6 inhibitor AS1517499 efficiently blocked IL-4 treatment-induced STAT6-pY641 in MHH-ES-1 cells (Fig. 6f and Supplementary Fig. 8c). AS1517499 treatment also efficiently suppressed increased STAT6-pY641, as well as Akt-pT308 and ERK-p42/p44 signals induced by chemotherapeutics including SN-38 (Fig. 6g) and etoposide (Fig. 6h), further supporting an essential role of increased STAT6

transcriptional activity in governing chemotherapy-induced activation of Akt and ERK.

Notably, SN-38 treatment increased cellular STAT6-pY641 signals within 2 h (Fig. 6e). This rapid activation suggests that STAT6 phosphorylation may be largely due to activation of JAKs, rather than secretion of ILs[39]. Considering the lack of known target(s) for AS1517499 since this compound was screened as a STAT6 transcriptional activity inhibitor in response to IL-4 stimulation[40], and JAK1 has been shown to be responsible for STAT6-pY641[41], we then tested whether specific JAK1 inhibitors blocked chemotherapy-induced STAT6-pY641 and activation of Akt/ERK. To this end, we focused on filgotinib, a small molecule JAK1 inhibitor initially identified in cellular assays[42] and has been approved for treating rheumatoid arthritis patients in Europe and Japan with ongoing clinical trials largely in the US (such as NCT05090410 and reviewed in ref. 43). We first identified a concentration of filgotinib that suppresses IL-4-induced STAT6-pY641 signals in MHH-ES-1 cells (Fig. 6i and Supplementary Fig. 8d). We then found that filgotinib at this concentration was sufficient to block SN-38 (Fig. 6j) or etoposide (Fig. 6k) -triggered STAT6-pY641 and subsequently increased Akt-pT308 and ERK-p42/p44 signals. Consistently, filgotinib also blocked SN-38-induced GAS6 transcription in MHH-ES-1 cells (Supplementary Fig. 8e). These data suggest that chemotherapeutics induce JAK1 activation in Ewing sarcoma to phosphorylate STAT6-Y641, triggering its transcriptional activation of GAS6, which activates TAM/Akt/ERK signaling and enhances chemoresistance in an autocrine or/and paracrine manner.

Considering previously STAT6 Ser phosphorylation by TBK1[44], ERK[45], or JNK[46] has also been reported to differentially regulate STAT6 transcriptional activity in addition to STAT6 Tyr phosphorylation, and chemotherapy-induced ERK activation in Ewing sarcoma (Fig. 1), we also examined if chemo-induced ERK activation regulates chemoresistance through STAT6 Ser phosphorylation. To this end, we found that ERK inhibition by PD0325901 failed to reduce SN-38-induced GAS6 transcription in MHH-ES-1 cells (Supplementary Fig. 8f), suggesting ERK-mediated STAT6 Ser phosphorylation may not play a critical role here. Consistently, PD0325901 didn't affect SN-38-induced STAT6-pY641 and Akt-pT308 in MHH-ES-1 cells (Supplementary Fig. 8g). These data suggest ERK/STAT6 signaling might not contribute to Ewing sarcoma chemoresistance.

Previously, GAS6 was also reported to promote STAT3 activation[47] and STAT3 was observed to contribute to chemoresistance in multiple cancer types[48]. To examine if STAT3 plays a role in Ewing sarcoma chemoresistance, we depleted endogenous STAT3 in MHH-ES-1 cells and found STAT3 depletion didn't affect SN-38-induced Akt-pT308 and ERK-p42/p44 (Supplementary Fig. 8h, i). Moreover, STAT3 depletion didn't significantly affect MHH-ES-1 cell responses to SN-38 treatment in vitro (Supplementary Fig. 8j). These data suggest that in Ewing sarcoma, STAT3 is less likely to play a role in regulating chemotherapy responses.

### Chemotherapy induces JAK1 activation by DNA-damaging kinases-mediated JAK1-SQ phosphorylation

Canonical JAK1 activation is mediated by binding of JAK1 using its N-terminal FERM/SH2 domains to IL-receptors, which brings JAK kinases together to trigger its auto-phosphorylation and activation, subsequently recruiting STAT6 to phosphorylate STAT6-pY641[49]. We observed that neither SN-38 nor etoposide significantly induced JAK1-pY1034/pY1035 known to be triggered by ILs (Supplementary Fig. 9a–c). However, immunoprecipitated FL (full-length)-JAK1 kinases (Fig. 7a) from either SN-38 (Fig. 7b) or etoposide (Fig. 7c) treated MHH-ES-1 cells displayed an enhanced kinase activity towards phosphorylating bacterially purified GST-STAT6 proteins in in vitro kinase assays, suggesting these chemotherapeutic agents activate JAK1 in a cytokine-independent mechanism. Notably, chemotherapy-induced DNA damage triggers the activation of DNA-damaging kinases

including ATM, ATR, and DNAPK to repair damaged DNA[50], and all of them recognize a consensus SQ/TQ motif[51] for phosphorylation. We found that treatment with either SN-38 (Fig. 7d) or etoposide (Fig. 7e) induced JAK1-pSQ in cells. Querying the human JAK1 protein sequence identified only one S571Q motif located close to the JAK1-JH2 pseudokinase domain (Fig. 7a) and the S571 residue is structurally available for post-translational modifications (Supplementary Fig. 9d). Mutating S571 to Ala abolished SN-38 induced JAK1-pSQ signals in cells (Fig. 7f). As chemotherapy-induced activation of ATM, ATR, and DNAPK[52], we inhibited these kinases and found ATR inhibition (by VE-822) suppressed SN-38-induced JAK1-pSQ (Supplementary Fig. 9e), whereas inhibiting ATM (by NU-55933) or DNAPK (by NU-7441) attenuated etoposide-triggered JAK1-pSQ signals (Supplementary Fig. 9f) in cells. Notably, it seemed SN-38-induced JAK1-pS571Q largely occurred in the nucleus (Supplementary Fig. 9g), and JAK1-pS571Q didn't significantly affect JAK1 cellular localization (Supplementary Fig. 9h).

To examine if JAK1-S571Q phosphorylation regulates JAK1 kinase activity, we immunoprecipitated Flag-JAK1-KM (kinase motif, including both JH1 and JH2, Fig. 7a)-WT, S571A, S571D, and K908A (a kinase-dead mutant[53]) from cells and performed in vitro JAK1 kinase assays using recombinant GST-STAT6 proteins. We found the JAK1-S571A mutant displayed reduced kinase activity towards phosphorylating STAT6 compared with WT or S571D-JAK1 (Supplementary Fig. 9i). More importantly, we observed SN-38 treatment only enhanced WT-JAK1 activity but not S571A nor S571D-JAK1 mutants (Fig. 7g, h). These data suggest that chemotherapy may induce JAK1 kinase activation by phosphorylating the JAK1-S571Q motif through DNA damage-responsive kinases. Consistently, SN-38 promoted activation of Akt and ERK in a WT-JAK1 but not S571A-JAK1-dependent manner (Fig. 7i, j). As a result, compared with WT-JAK1 expressing MHH-ES-1 cells, S571A-JAK1 mutant expression sensitized cells to SN-38 treatment (Fig. 7k) due to the inability of this mutant to mediate SN-38-induced JAK1 activation.

We then explored the mechanism by which JAK1-pS571 promotes JAK1 activation. We found that SN-38 treatment promoted activation of a truncated JAK1-KM (lacking the N-terminal FERM and SH2 domains for binding IL-receptors) (Fig. 7l), suggesting chemotherapy-induced JAK1 activation does not require the FERM and SH2 domains. Previously it has been shown that the pseudokinase JH2 domain allosterically inhibits JAK kinase activity. For example, JH2 autoinhibits JH1 by binding and stabilizing the inactive JH1 tyrosine kinase domain in JAK2[54]. The SH2/JH2 linker was shown to enhance JH2 inhibitory effects on JH1 and reduce JH2 affinity with ATP[55]. We found that SN-38 treatment reduced JH2 binding to JH1-JAK1 in cells (Fig. 7m), supporting the role of JAK1-S571 phosphorylation in activating JAK1 via releasing JH2 pseudokinase domain inhibition on JH1 kinase domain. Thus, different from the canonical cytokine-induced "outside-in" JAK1 activation mechanism depending on the N-terminal FERM/SH2 domains, chemotherapy may activate JAK1 via an intracellular mechanism through JAK1-S571 phosphorylation by directly releasing JH2 suppression on JH1 (Fig. 7n).

### Filgotinib sensitizes Ewing sarcoma cells to chemotherapy in vitro and in vivo

We then tested whether, like TAM kinase inhibition, inhibiting JAK1/STAT6 signaling also enhances chemotherapy efficacy given our data suggests JAK1/STAT6 inhibition suppresses GAS6 expression necessary for TAM kinase activation. To this end, we observed that the STAT6 inhibitor AS1517499 sensitized MHH-ES-1 cells to either SN-38 (Fig. 8a) or etoposide (Fig. 8b) in vitro. Similarly, filgotinib enhanced either SN-38 (Fig. 8c, d) or etoposide (Fig. 8e, f)-induced MHH-ES-1 or TC-71 cell death in vitro. Moreover, we established MHH-ES-1 xenografts and treated these tumor-bearing mice with vehicle, filgotinib (10 mg/kg, IP), irinotecan (10 mg/kg, IP), and the combination. We found that the combination of filgotinib + irinotecan was more

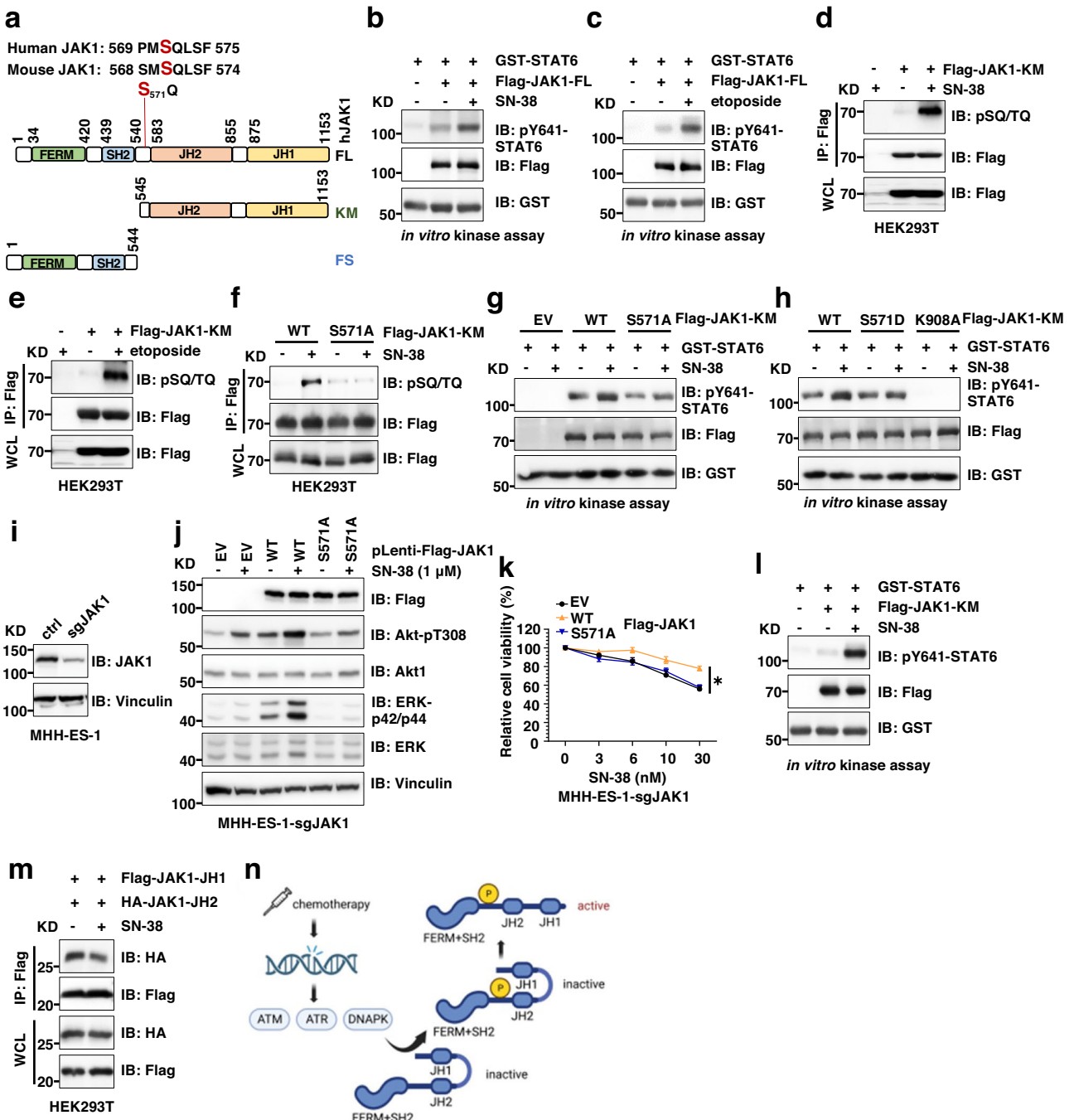

**Fig. 7 | Chemotherapy promotes JAK1-pSQ to promote JAK1 activation. a** A cartoon illustration of the human JAK1 domain structure with the only S571Q motif labeled. **b**, **c** IB analysis of in vitro JAK1 kinase assay reactions by incubating indicated Flag-immunoprecipitated JAK1-FL (full-length) from control or SN-38 (1 μM for 24 h) treated MHH-ES-1 cells with bacterially purified GST-STAT6 (aa 633–801) proteins. Refer to the Method section for details. **d**–**f** IB analysis of Flag-IPs and WCL from HEK293T cells transfected with indicated DNA constructs and treated with either SN-38 (1 μM) or etoposide (10 μM) for 24 h. **g**, **h** IB analysis of in vitro JAK1 kinase assay reactions by incubating indicated Flag-immunoprecipitated JAK1-KM (kinase motif) from control or SN-38 (1 μM for 24 h) treated MHH-ES-1 cells with bacterially purified GST-STAT6 (aa 633–801) proteins. Refer to the Method section for details. **i** IB analysis of indicated MHH-ES-1 cells. **j** IB analysis of WCL from MHH-ES-1 cells depleted of endogenous JAK1 stably expressing indicated Flag-JAK1 treated with 1 μM SN–38 for 24 h. **k** Cell viability assays using indicated MHH-ES-1 cells treated with indicated doses of SN-38 for 48 h. Error bars were calculated as mean ± SD, $n = 3$ (experimental triplicates). *$p < 0.05$ (one-way ANOVA test). *$p$ value of each point of **k**: WT/EV: <0.0001 (30 nM), WT/S571A: <0.0001 (30 nM). **l** IB analysis of in vitro JAK1 kinase assay reactions by incubating indicated Flag-immunoprecipitated JAK1-KM from control or SN-38 (1 μM for 24 h) treated MHH-ES-1 cells with bacterially purified GST-STAT6 (aa 633–801) proteins. Refer to the Method section for details. **m** IB analysis of Flag-IPs and WCL from HEK293T cells transfected with indicated DNA constructs and treated with either SN-38 (1 μM) for 24 h. **n** A cartoon model revealing chemotherapy induces DNA damage, leading to activation of DNA-damaging kinases including ATM, ATR, and DNAPK that subsequently phosphorylates JAK1 at S571Q motif to activate JAK1 intracellularly via releasing JAK1-JH2 mediated suppression of its JH1 kinase domain. WB data presented in this figure are representative data from biological triplicates.

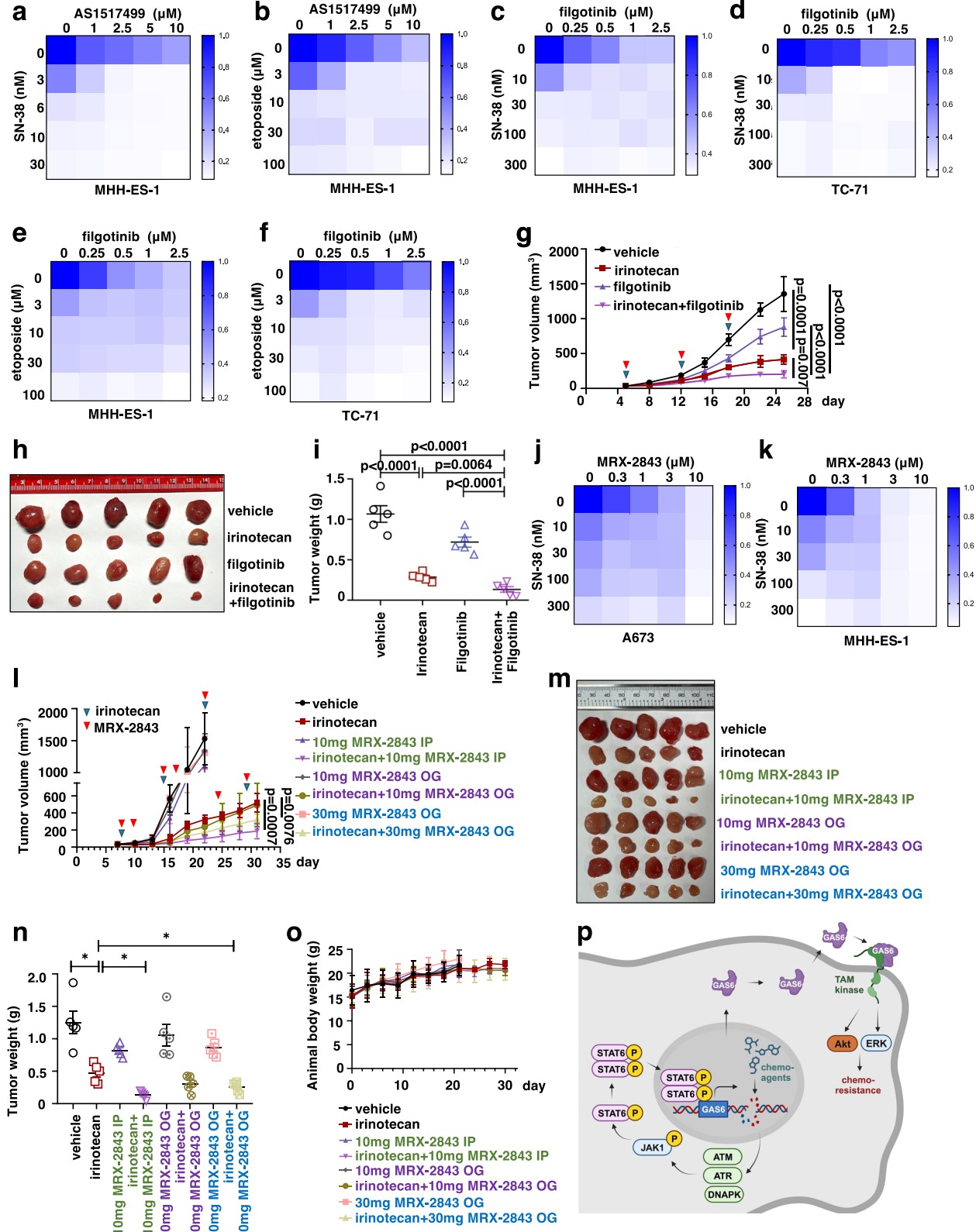

effective in suppressing xenograft tumor growth compared with each of the single-agent treatments (Fig. 8g–i) with no obvious effects on animal weights (Supplementary Fig. 10a). Given that filgotinib is currently used in Europe and Japan for treating rheumatoid arthritis patients with ongoing US clinical (such as NCT05090410 and reviewed in ref. 43), our data justify a clinical study to evaluate the combinatory effects of filgotinib + irinotecan in treating Ewing sarcoma patients.

## The clinical compound MRX-2843 enhances irinotecan effects to further suppress Ewing sarcoma tumor growth

Given the closest analog of UNC2025 is UNC2371, now currently a clinical candidate named MRX-2843 as an oral agent for advanced (i) solid tumors, (ii) osimertinib-resistant non-small cell lung cancer, and (iii) adolescents/adults with refractory AML/ALL or mixed phenotype acute leukemia, we examined MRX-2843 effects in Ewing sarcoma cell

**Fig. 8 | Filgotinib or MRX-2843 sensitizes Ewing sarcoma to chemotherapy in vitro and in vivo. a–f** Representative heatmaps for cell viability using MHH-ES-1 (**a**, **b**, **c**, **e**) or TC-71 cells (**d**, **f**) treated with indicated doses of compounds for 2 days. **g** Tumor volume measurements at indicated days post-injection of MHH-ES-1 cells subcutaneously with indicated compounds (irinotecan (10 mg/kg) and filgotinib (10 mg/kg)) injected by IP. Error bars were calculated as mean ± SD, $n = 5$ tumors. $p$ values are calculated by two-way ANOVA followed by the Tukey multiple comparison test. **h** Isolated tumors from (**g**) and weighed in (**i**). Error bars were calculated as mean ± SD, $n = 5$ tumors. $p$ values are calculated by a two-tailed student's $t$-test. **j**, **k** Representative heatmaps for cell viability in MHH-ES-1 cells treated with indicated doses of compounds for 2 days. **l** Tumor volume measurements at indicated days post-injection of MHH-ES-1 cells with indicated compounds IP injected or by oral gavage at the indicated days. Error bars were calculated as mean ± SD, $n = 5$

tumors. $p$ values are calculated as indicated (two-way ANOVA followed by Tukey multiple comparison test). **m** Isolated tumors from (**l**) and weighed in (**n**). Error bars were calculated as mean ± SD, $n = 5$ tumors. $p$ values are calculated as indicated (two-tailed student's $t$- test). **o** Animal body weight measurements at indicated days. Error bars were calculated as mean ± SD, $n = 5$ animals in each group. **p** A cartoon illustration for a proposed model. Chemotherapy triggers DNA damage and activates DNA-damaging kinases to phosphorylate the JAK1-S571Q motif leading to JAK1 activation and subsequent STAT6-Y641 phosphorylation. This causes STAT6 nuclear translocation to promote GAS6 transcription and secretion that binds and activates TAM kinases via either autocrine or paracrine manners, which further activates Akt and ERK signaling contributing to chemoresistance. Blocking either JAK1 by filgotinib or TAM kinases by clinical inhibitor MRX-2843 sensitizes Ewing sarcoma to conventional chemotherapy.

responses to chemotherapy. Similar to UNC2025 (Fig. 3a, b), MRX-2843 sensitized both A673 (Fig. 8j) and MHH-ES-1 (Fig. 8k) cells to SN-38. More importantly, we found in xenografted MHH-ES-1, twice a week administration of MRX-2843 (10 mg/kg) by IP similarly exerted an additive effect with irinotecan to reduce tumor growth (Fig. 8l–n) as UNC2025 did (Fig. 3i). Administration of MRX-2843 via oral gavage twice weekly achieved a combination effect with irinotecan at a dose of 30 mg/kg but not at 10 mg/kg (Fig. 8l–n). This suggests that the route of drug administration affects combination therapeutic effects. Notably, compared with previous preclinical studies using MRX-2843 at a dose of 50 mg/kg with daily oral gavage administration[56], the reduced dose/frequency of MRX-2843 administration in our animal studies retained its activity. Nonetheless, neither of these treatments affected animal health and body weight during the experiment (Fig. 8o), suggesting these treatments are well-tolerated by animals. In addition, to better mimic clinical settings, we allowed the xenografted MHH-ES-1 tumors grown to at least 250 mm³, followed by single treatments with either irinotecan or MRX-2843, or the combination. We found the combination treatment more significantly reduced established large tumor growth in mice compared with each of the single treatments (Supplementary Fig. 10b–d) with neglectable toxicity (Supplementary Fig. 10e). Moreover, we also tested the efficacy of this combination therapy in Ewing sarcoma PDX models. We xenografted freshly prepared NCH-EWS-1 PDX tumor cells subcutaneously into nude mice and started treatments when tumors are noticeable. We found either irinotecan alone (10 mg/kg, IP) or MRX-2843 alone (10 mg/kg, IP) reduced NCH-EWS-1 PDX tumor growth, while the combination further significantly reduced PDX tumor growth (Supplementary Fig. 10f–h). Notably, these treatments were well-tolerated by animals (Supplementary Fig. 10i). These data support the clinical compound MRX-2843 enhances irinotecan-mediated tumor growth suppression effects in Ewing sarcoma murine models derived from either Ewing sarcoma cells or Ewing sarcoma PDX tumors, supporting a further clinical investigation to evaluate their treatment efficacy.

## Discussion

As a rare pediatric cancer with an annual incidence of ~200 cases in the United States, treatment for Ewing sarcoma patients have been limited to cytotoxic chemotherapy, radiation, and surgery. Chemo-toxicity and chemoresistance have been major clinical problems for Ewing sarcoma patients. Efforts have been devoted to identify combination therapeutic options to alleviate issues associated with chemotherapy; however, the recent failure of IGF1R-targeted therapy in synergizing with chemotherapy to improve treatment outcomes in clinical trials[19] speaks to the need for novel therapeutic directions. The lack of molecular understanding of chemotherapy resistance mechanisms hinders the development of novel chemo-sensitization therapies. Using phospho-profiling, genetic, and biochemical approaches, our studies unravel a chemotherapy-induced mechanism for JAK1/STAT6/GAS6-mediated autocrine or paracrine signaling contributing to Ewing sarcoma chemoresistance, by activating TAM kinases and subsequent

downstream Akt and ERK signaling. Chemotherapy-induced DNA damage triggers JAK1-S571Q phosphorylation, a site proximal to the JAK1-JH2 pseudokinase domain. Phosphorylation releases JH2-mediated inhibition of the JAK1-JH1 tyrosine kinase domain, leading to JAK1 kinase activation and subsequent STAT6 phosphorylation. This alternative JAK1 activation mechanism initiated from inside of cells is distinct from the canonical IL-receptor mediated JAK1 activation largely controlled by JAK1 N-terminal FERM/SH2 domains through an "outside-in" sensing mechanism[49]. However, our study does not rule out the possibility that JAK1-pSQ precludes binding of JAK1 negative regulators such as SOCS1/3, which warrants further investigations. In addition, our studies also define chemotherapy or DNA damage as another upstream pathophysiological cue for JAK1 activation. Considering the JAK family of kinases, including JAK1, JAK2, JAK3, and TYK2 share a similar domain structure and activation mechanisms, it is plausible that other JAK kinase members are also similarly regulated by DNA damage for kinase activity control. Moreover, most JAK kinases have been shown to play critical roles in immune regulation either in immune cells or tumor cells[57], warranting further in-depth investigations to examine if DNA damage-induced JAK kinase activation modulates the tumor immune microenvironment. Since there is no syngenetic murine models have ever been successfully developed to faithfully mimic Ewing sarcoma, this test will need to be performed in other defined tumor types with available genetic disease murine models.

As members of the family of vitamin K-dependent proteins, GAS6 is a protein ligand for all three TAM RTKs, while PROS1 is a protein ligand for MERTK and TYRO3. The PtdSer, GAS6/AXL, and GAS6/MERTK signaling have been extensively studied both as a tumor survival, chemoresistance factor in the tumor cell itself where the RTKs are often overexpressed and as stimulating immunosuppression mechanisms from the normal myeloid cells infiltrating the tumor microenvironment (see refs. 31,32,58–60 for reviews). Signaling initiates efferocytosis to clear apoptotic tumor cells[61] and facilitates tumor survival, invasion, and angiogenesis[62,63]. Less is known about the control of GAS6 and PROS1 although both have been observed in the tumor microenvironment released from both tumor cells and infiltrating immune cells. Reduced GAS6 expression was observed in advanced breast cancer[64] while inhibiting the GAS6/AXL signaling suppressed gastric cancer progression[65]. In addition, previous studies reported that stimulation of the GAS6/AXL signaling recruits Src and focal adhesion kinase (FAK), which in turn leads to enhanced proliferation of schwannoma[66]. Blocking GAS6/AXL signaling was also reported to increase DNA damage to sensitize ovarian cancer to chemotherapy and PARP inhibition[67]. However, the pathophysiological regulatory cues for GAS6 expression are less well studied. It has been reported that hypoxia-stabilized HIF transcription factor promotes GAS6 transcription[68], estrogen-induced GAS6 transcription in breast cancer[34], and IL4/STAT6 signaling governed GAS6 transcription in macrophages[68]. Our study reveals the chemo/JAK1/STAT6 signaling also promotes GAS6 transcription and secretion in Ewing sarcoma (Fig. 8p).

A series of ATP-binding site-directed MERTK kinase inhibitors have been developed at UNC with varying degrees of selectivity and drug-like properties including oral bioavailability. These include UNC2025[27], UNC5293[28], and UNC2371[56]. Each is relatively selective for MERTK with Kis in the 0.14–0.19 nM range but UNC2025 and UNC2371 with a potency for AXL (Kis: 4.2 and 13.3 nM, respectively) that is like other potent AXL small molecule inhibitors, while UNC5293 is much weaker against AXL with 97.5 nM Ki. UNC2371 has been advanced as the clinical candidate by a UNC start-up, Meryx, as MRX-2843. This compound is now in Phase 1 clinical trials in (i) advanced solid tumors (trial# NCT03510104), (ii) osimertinib-resistant NSCLC (trial# NCT04762199) and relapsed AML in adults and adolescents (trial# NCT04872478). Our preclinical testing results indicated that either UNC2025 or the clinical compound MRX-2843 sensitized Ewing sarcoma cells or Ewing sarcoma tumors to chemotherapy in vitro or in vivo, respectively. Our data provide a strong rationale for promising combinations rapidly assessed to evaluate the potential for additional benefit from TAM kinase inhibition in the context of chemotherapy.

Although chemotherapeutic agents induced activation of Akt/ERK in Ewing sarcoma, it does not necessarily prove activated Akt/ERK signaling contributes to chemoresistance. Due to the rareness of this disease, limited clinical resources have been obtained to systematically investigate potential chemo-resistant mechanisms from patient cohorts. Interestingly, a recent study determined the transcriptome changes in four Ewing sarcoma patients with good responses to chemotherapy vs. nine patients with poor responses[69]. Re-analyzing the published data led us to find that increased JAK1 and STAT6 expression were identified in these patients as positively correlated with poorer response to chemotherapy (Supplementary Fig. 10j). However, due to the small number of patients in this study, we cannot conclude if increased JAK1/STAT6 expression contributes to Ewing sarcoma chemoresistance. Our goal in this study is to improve chemotherapy efficacy instead of overcoming chemoresistance. We are of the opinion that Ewing sarcoma tumors may increase Akt/ERK signaling upon chemotherapy to provide cell survival protection, a period allowing Ewing sarcoma tumors to develop stable resistance mechanisms such as genetic changes. Thus, if we shorten this protection window by inhibiting JAK/TAM/GAS6 signaling as in this study, this improves chemotherapy efficacy.

Notably, few genetic mutations including p53 mutations were observed in Ewing sarcoma patients, previous studies strongly support p53/STAG2 mutations associate with a metastatic phenotype[70–72] with a worse prognosis[73]. A673 cells bear a p53-A119fs and MHH-ES-1 cells contain a p53 S215del and STAG2-Q375fs. If mutations of p53 and/or STAG2 contribute to chemoresistance or chemosensitivity remain to be determined. Considering chemoresistance can also be caused by genetic changes[74], epigenetic changes[75], and metabolic changes[76], identification of additional chemoresistance mechanisms are warranted for future studies.

In addition to Ewing sarcoma, adjuvant chemotherapy remains a standard-of-care for many cancers, e.g., breast (TNBC), colorectal, pancreatic, lung, and ovarian cancer. Our identified chemotherapy-induced JAK1/STAT6/GAS6/TAM signaling contribution to chemoresistance in cancer warrants study beyond Ewing sarcoma. Whether JAK1 or TAM kinase inhibition similarly sensitizes Ewing sarcoma to radiation which also relies on generating DNA damage remains to be determined. Nonetheless, considering the close relationship of either JAK1 inhibitors or TAM kinase inhibitors to abrogating resistance or lowering chemotherapy doses merits further investigations.

## Methods
### Cell culture and transfection
Human Ewing sarcoma cell line A673, MHH-ES-1, and SK-N-MC, human immortalized kidney cell lines HEK293T were cultured in DMEM medium supplemented with 10% FBS, 100U penicillin and 100 mg/mL streptomycin unless otherwise stated. Human Ewing sarcoma cell line EWS894 was cultured in RPMI-1640 medium supplemented with 10% FBS, 100U penicillin, and 100 mg/mL streptomycin unless otherwise stated. Human Ewing sarcoma cell lines TC-32, TC-71, CHLA-10, and CHLA-32 were cultured in IMDM medium supplemented with 20% FBS, 100U penicillin, and 100 mg/mL streptomycin unless otherwise stated. Ewing sarcoma cells used in this study were obtained either from Dr. Ian Davis lab at UNC or COG (children's oncology group). Cells have been monitored for mycoplasma contamination. MHH-ES-1, A673, and SK-N-MC cells have been recently authenticated by the Davis Lab by STR analyses (Genetica). MHH-ES-1 and TC-71 cells were further authenticated by the Liu Lab by STR analyses (Genetica). Other cell lines used in this study were verified by the manufacturer's websites and regularly checked by morphology for authentication. The Ewing sarcoma PDX tumor NCH-EWS-1 was obtained from Dr. Peter J. Houghton at UT Health San Antonio.

Cell transfection was performed using polyethylenimine (PEI), as described previously[16,77]. The packaging of lentiviral shRNA or cDNA expressing viruses and subsequent infection of various cell lines followed the protocols described previously[77,78]. Following viral infection, cells were cultured in the presence of puromycin (1 μg/mL) depending on the viral vector used to infect cells.

### Reagents
Puromycin (P8833), anti-HA agarose beads (A-2095), and anti-Flag agarose beads (A-2220) were purchased from Sigma. Irinotecan (I0714), etoposide (E0675), and Temozolomide (T2744) were purchased from TCI. Doxorubicin (BP2516-5) and Blasticidin (15205) were purchased from Fisher Bioreagent. Polyethylenimine (PEI) (23866-1) was purchased from Polysciences, Inc. IL-4 was purchased from R&D Systems (204-IL-010). SN-38 (HY-13704), AS1517499 (HY-100614) and filgotinib (HY-18300) were purchased from MedChemExpress. UNC2025, UC5293, UNC5567, and MRX-2843 were produced by the UNC drug discovery unit. MK2206 (S2808), cabozantinib (S1119), dasatinib (S1021), pazopanib (S3012), sunitinib (S7781), BKM120 (S2247), Torin 2 (S2817), IPI-549 (S8330), rapamycin (S1039), S6K-18 (S0385), PD0325901 (S1036), and filgotinib (S7605) were purchased from Selleck.

### Antibodies
All antibodies were used at a 1:1000 dilution in TBST buffer with 5% non-fat milk for western blotting. Anti-pT308-AKT antibody (2965), anti-AKT1 antibody (2938), anti-GST antibody (2625), anti-p42/44-ERK (4370), Anti-HA antibody (3724), GAS6 (67202), c-PARP (5625), c-caspase 3 (672029661), STAT6 (9362), STAT6-pY641 (9364), TEAD1 (12292), pSQ/TQ (2851), STAT3 (9139), JAK1-pY1034/1035 (74129), JAK1 (50996), anti-rabbit IgG, HRP-linked antibody (7074), and anti-mouse IgG, HRP-linked antibody (7076), were obtained from Cell Signaling Technology. anti-ERK (sc-135900), anti-AXL antibody (sc-166268), anti-TYRO3 antibody (sc-166359), Anti-GST antibody (sc-138), Lamin A/C (sc-7293), and anti-vinculin antibody (sc-25336) were obtained from Santa Cruz Biotechnology. Polyclonal anti-Flag antibody (F-7425), monoclonal anti-Flag antibody (F-3165, clone M2) were obtained from Sigma. anti-Flag agarose beads (A-2220), anti-HA agarose beads (A-2095) were obtained from Sigma. anti-Tubulin antibody (10068-1-AP) were obtained from Proteintech. Anti-MERTK antibody was generated by the Earp lab.

### Plasmids
HA-STAT6 was cloned into pcDNA3-HA vector using EcoRI and XhoI enzyme sites. pLenti-HA-STAT6 was cloned into the pLenti-HA-blasticidin vector using XbaI and AgeI enzyme sites. GST-STAT6-truncation (aa 633–801) were cloned into the pGEX-6p-1 vector using BamHI and SalI enzyme sites. Flag-JAK1-FL, KM, FS, JH1, HA-JAK1-KM, and JH2 were cloned into pLenti-GFP-blasticidin vector using XbaI and

SalI enzyme sites. S571A-KM, S571D-KM, and K908A-KM related constructs were obtained using Site-Directed Mutagenesis Kits (Agilent 200523).

The primers used for plasmid construction in this study are listed below:

STAT6-EcoRI-F: 5′-GCATGAATTCTCTCTGTGGGGTCTGG-3′
STAT6-XhoI-R: 5′-GCATCTCGAGTCACCAACTGGGGGTTGG-3′
STAT6-XbaI-F: 5′-GCATTCTAGATCTCTGTGGGGTCTGG-3′
STAT6-AgeI-R: 5′-GCATACCGGTTCACCAACTGGGGGTTGG-3′
GST-STAT6-633-BamHI-F: 5′-GCATGATCCCAGATGGGTAAGGATG GCAG-3′
GST-STAT6-801-SalI-R: 5′-GCATTCAGTCGACCTGTTCAGTGGGAG GC-3′
JAK1-XbaI-Flag-F: 5′-GCATTCTAGAATGGATTACAAAGACGATGAC GATAAGCAGTATCTAAATATAAAAGAGGACTGC-3′
JAK1-SalI-R: 5′-GCATGTCGACTTATTTTAAAAGTGCTTCAAATCCT TCAATAAGG-3′
JAK1-S571A-F: 5′-GTCTACCCCATGGCCCAGCTGAGTTTCGATCGG-3′
JAK1-S571A-R: 5′-CCGATCGAAACTCAGCTGGGCCATGGGGTAGA C-3′
JAK1-S571D-F: 5′-GTCTACCCCATGGACCAGCTGAGTTTCGATCGG-3′
JAK1-S571D-R: 5′-CCGATCGAAACTCAGCTGGTCCATGGGGTAGA C-3′
JAK1-K908A-F: 5′-GAGCAGGTGGCTGTTGCATCTCTGAAGCCTGA GAG-3′
JAK1-K908A-R: 5′-CTCTCAGGCTTCAGAGATGCAACAGCCACCTG CTC-3′
JAK1-XbaI-HA-545-F: 5′-GCATTCTAGAATGTACCCATACGATGTTC CAGATTACGCTCAGCCCAAGCCCCGA-3′
JAK1-SalI-855-R: 5′-GCATGTCGACTTAAATATCTGGATTCTGCTCT TCAAG-3′
JAK1-XbaI-856-F:5′-GCATTCTAGAATGGATTACAAAGACGATGAC GATAAGGTTTCAGAAAAAAAACCAGCAACTGAAG-3′

The sequence for sgRNAs for CRISPR-Cas9 mediated deletion of the indicated genes were generated by cloning the annealed sgRNAs into a BsmBI-digested pLenti-CRISPRv2 vector (Addgene 52961). The short guide RNAs (sgRNA) were designed based on predictions from crispr.mit.edu and oligonucleotides were obtained from Eton Biosciences. CRISPR sgRNAs were designed as listed below:

sgControl-F: 5′-CACCGCTTGTTGCGTATACGAGACT-3′
sgControl-R: 5′-AAACAGTCTCGTATACGCAACAAG-3′
MERTK-sg1-F: 5′-CACCGGTAATTTCTCTCCGGACGGA-3′
MERTK-sg1-R: 5′-AAACTCCGTCCGGAGAGAAATTACC-3′
MERTK-sg2-F: 5′-CACCGCCCGGGAATAGCGGGTAAGG-3′
MERTK-sg2-R: 5′-AAACCCTTACCCGCTATTCCCGGGC-3′
AXL-sg4-F: 5′-CACCGGAGAGCCCCCCGAGGTACAT-3′
AXL-sg4-R: 5′-AAACATGTACCTCGGGGGGCTCTCC-3′
AXL-sg5-F: 5′-CACCGCGGGCACCTGTGATATTCCC-3′
AXL-sg5-R: 5′-AAACTGGCGTTATGGGCTTCGCAGC-3′
TYRO3-sg3-F: 5′-CACCGCCCTTTCCAACTGTCTTGTG-3′
TYRO3-sg3-R: 5′-AAACCACAAGACAGTTGGAAAGGGC-3′
TYRO3-sg5-F: 5′-CACCGTGTGAAGCTCACAACCTAAA-3′
TYRO3-sg5-R: 5′-AAACTTTAGGTTGTGAGCTTCACAC-3′
IGF1R-sg1-F: 5′-CACCGGTGGAGAACGACCATATCCG-3′
IGF1R-sg1-R: 5′-AAACCGGATATGGTCGTTCTCCACC-3′
IGF1R-sg2-F: 5′-CACCGCCACGACGGCGAGTGCATGC-3′
IGF1R-sg2-R: 5′-AAACGCATGCACTCGCCGTCGTGGC-3′
IGF1R-sg3-F: 5′-CACCGGGCTCTCTCCCCGTTGTTCC-3′
IGF1R-sg3-R: 5′-AAACGGAACAACGGGGAGAGAGCCC-3′
GAS6-sg1-F: 5′-CACCGAACTGCGTGGCCTCGCGCGC-3′
GAS6-sg1-R: 5′-AAACGCGCGCGAGGCCACGCAGTTC-3′
GAS6-sg2-F: 5′-CACCGCCAGGACATGGACACCTGTG-3′
GAS6-sg2-R: 5′-AAACCACAGGTGTCCATGTCCTGGC-3′
STAT6-sg1-F: 5′-CACCGCCGGGGAATACCTGGTGACG-3′

STAT6-sg1-R: 5′-AAACCGTCACCAGGTATTCCCCGGC-3′
STAT6-sg2-F: 5′-CACCGTCCTGAGAACCCTCGTCACC-3′
STAT6-sg2-R: 5′-AAACGGTGACGAGGGTTCTCAGGAC-3′
NF-Y-sg1-F: 5′-CACCGTACGTTACCTTTCCCGTTTG-3′
NF-Y-sg1-R: 5′-AAACCAAACGGGAAAGGTAACGTAC-3′
NF-Y-sg2-F: 5′-CACCGTTACACAACATCATATCAAC-3′
NF-Y-sg2-R: 5′-AAACGTTGATATGATGTTGTGTAAC-3′
STAT3-sg1-F: 5′-CACCGCTGCTGCTTCTCCGTCACCA-3′
STAT3-sg1-R: 5′-AAACTGGTGACGGAGAAGCAGCAGC-3′
STAT3-sg2-F: 5′-CACCGAGATTGCCCGGATTGTGGCC-3′
STAT3-sg2-R: 5′-AAACGGCCACAATCCGGGCAATCTC-3′
JAK1-sg-F: 5′-CACCGTTGATGACAAGATGTCCCTC-3′
JAK1-sg-R: 5′-AAACGAGGGACATCTTGTCATCAAC-3′

## Phospho-profiling
MHH-ES-1 cells treated with DMSO or SN-38 (1 μM) for 24 h were lysed with Triton buffer (50 mM Tris pH 7.5, 120 mM NaCl, 1% Triton X-100) supplemented with protease inhibitor cocktail and phosphatase inhibitor cocktail. About 2 mg of WCL from each group was applied to the Proteome Profiler Human Phospho-Kinase Array Kit (Biotechne #ARY003C) following the manufacturer's instructions. Phospho-signals were obtained by standard immunoblot analyses.

## Immunoblot and immunoprecipitation analyses
Cell lysis was performed using EBC buffer (50 mM Tris pH 7.5, 120 mM NaCl, 0.5% NP-40) or Triton buffer (50 mM Tris pH 7.5, 120 mM NaCl, 1% Triton X-100) supplemented with protease inhibitor cocktail and phosphatase inhibitor cocktail. The protein concentrations of whole cell lysates were determined using Bio-Rad protein assay reagents and NanoDrop OneC as described previously[16,77]. Equal amounts of whole cell lysates were loaded by SDS-PAGE and immunoblotted with indicated antibodies. For GST pulldown and immunoprecipitations analysis, 1 mg total lysates were incubated with the indicated beads for 3–4 h at 4 °C. The recovered immunocomplexes were washed three times with NETN buffer (20 mM Tris, pH 8.0, 100 mM NaCl, 1 mM EDTA, and 0.5% NP-40), resolved by SDS-PAGE, and then immunoblotted with the indicated antibodies. For GST pulldown and immunoprecipitation analysis, 1 mg of total lysate was incubated with the designated beads for 3–4 h at 4 °C. The immunocomplexes obtained were washed three times with NETN buffer (20 mM Tris, pH 8.0, 100 mM NaCl, 1 mM EDTA, and 0.5% NP-40), resolved by SDS-PAGE, and then immunoblotted with the indicated antibodies. As indicated in the corresponding figures, quantification of western blot densitometry was performed using ImageJ to normalize indicated phospho-signals to total protein signal intensities.

## TCA (trichloroacetic acid) precipitation
Add 100 μl of 100% TCA to 1 ml of cell culture medium supernatant and vortex, precipitate on ice for 30 min. Centrifuge at 10,000×*g* for 15 min at 4 °C. Quickly and carefully aspirate the supernatant and wash the pellet once more with 500 μL of ice-cold acetone to remove any residual TCA, followed by centrifugation at 10,000×*g* for 5 min at 4 °C and aspiration of the supernatant. Allow the pellet to dry. Resuspend the sample in 50 μl of 2x SDS buffer. The samples were heated at 95 °C for 5 min and analyzed by SDS-PAGE.

## RNA extraction and qRT-PCR
RNA extraction was performed with an RNA miniprep super kit. The final elution step was done with 50 μL of RNAse-free water. The relative enrichment of mRNA was quantified with the NanoDrop OneC (Thermo Fisher Scientific). At least two experimental replicates were performed for RNA extraction. Reverse transcription was performed with an iScript cDNA synthesis kit. Quantitative real-time PCR was performed with iTaq universal SYBR green supermix using a

QuantStudio 6 Flex Real-Time PCR Systems (Thermo Fisher Scientific). Each mRNA level was normalized RNA18S to U6 snRNA. The comparative Ct method was used to calculate fold change in expression. Statistical significance was determined by one-way ANOVA tests. RT-PCR primers for the detection of GAS6 mRNAs are as below:

hGAS6-qPCR-F: 5′-GAACTTGCCAGGCTCCTACTCT-3′
hGAS6-qPCR-R: 5′-GGAGTTGACACAGGTCTGCTCA-3′
hPROS1-qPCR-F: 5′-TGGCAAGGAGACAGGTGTCAGT-3′
hPROS1-qPCR-R: 5′-GAGCAGTGGTAACTTCCAGGAG-3′
U6-qPCR-F: 5′-CTCGCTTCGGCAGCACA-3′
U6-qPCR-R: 5′-AACGCTTCACGAATTTGCGT-3′

## STAT6 cut&run-coupled PCR analyses

STAT6 cut&run experiment was performed according to the manufacturer's instructions (CST #86652). Briefly, MHH-ES-1 cells were treated with DMSO or 1 μM SN-38 for 24 h prior to cell collection. Experimental replicates were collected by trypsinization and 3x wash by sterile PBS. Suspend cell pellets in 1 mL of 1x wash buffer +spermidine+ protease inhibitor cocktail (PIC) by gentle pipetting. Cell pellets were washed three times with the same buffer. About 10 μL of activated Concanavalin A magnetic beads were incubated with 100 μL of prepared cells at room temperature for 15 min. Magnetic beads were enriched by a magnetic stand and the liquid was discarded. 100 μL of antibody binding buffer + spermidine + PIC was added then 1 μL of anti-STAT6 antibody (CST #5397) was added. Mixtures were transferred to 8-strip PCR tubes and rotated gently at 4 °C overnight. Mixtures were cleared on a magnetic stand and the liquid was removed. About 1 mL digitonin buffer (+spermidine + PIC) was used to wash beads, followed by the addition of 50 μL of pAG-MNase premix with 4 °C incubation for 1 h. Mixtures were cleared on a magnetic stand and beads were washed three times with digitonin buffer. About 150 μL of digitonin buffer was added to each tube and placed on ice for 5 min. About 3-μL cold calcium chloride was used to activate pAG-MNase and samples were incubated at 4 °C for 30 min. About 150 μL stop buffer was added and samples were incubated at 30 °C for 10 min to release DNA into solution. DNA was further purified by PCR product cleanup kits. About 10 μL of resulting DNA was used in end-point PCR as a template to amplify indicated SB1, SB2, SB3, or SB4 regions. PCR products were resolved on 1.5% TAE gels and imaged.

GAS6-qPCR-SB1-F: 5′-CCCCAGTGGGATTGGATCTG-3′
GAS6-qPCR-SB1-R: 5′-CCCCCGAATCTACTGCATCC-3′
GAS6-qPCR-SB2-F: 5′-TGGTCTCTGAAGACAAGCACA-3′
GAS6-qPCR-SB2-R: 5′-CGAGTGAAATGCGACGGTTT-3′
GAS6-qPCR-SB3-F: 5′-AAGTAGCCGTGGTGGTTTCG-3′
GAS6-qPCR-SB3-R: 5′-AACCAGAATGACGAGGCACT-3′
GAS6-qPCR-SB4-F: 5′-TCTGAGAATGGCAAGAACTCCA-3′
GAS6-qPCR-SB4-R: 5′-GTTCCAGCAGCCCATGGATA-3′

## Colony formation assays

Indicated cells were seeded into six-well plates (500 cells/well) and cultured in 37 °C incubator with 5% $CO_2$ for 2 weeks (as indicated in figure legends) until formation of visible colonies. Colonies were washed with 1xPBS, fixed with 75% ethanol for 5 min and stained with 0.5% crystal violet for 30 min. Colonies were then washed by distilled water and air-dried. Colony numbers were manually counted. At least two independent experiments were performed to generate the error bars.

## Immunofluorescence

Cells plated onto glass coverslips were fixed with 4% paraformaldehyde in PBS for 20 min at room temperature and permeabilized with 0.2% Triton X-100 for 20 min at room temperature. Cells were incubated with blocking buffer (5% bovine serum albumin and 0.1% Triton X-100 in PBS) for 1 h, incubated with primary antibodies at 4 °C overnight, incubated with secondary antibodies at room temperature for

1 h, and mounted with ProLong Gold antifade reagent. Fluorescent signals were observed with the Keyence BZ-X700 microscope.

## Immunohistochemistry

Tumor blocks were prepared by promptly fixing fresh xenograft tumors in 10% formalin for a duration of 2 days, followed by a transfer to 80% ethanol for an additional day. Subsequently, a 4 μm section was obtained from each tumor block by the UNC TPL facility for further IHC studies. IHC was performed as described previously[16]. Briefly, tumor sections were incubated with blocking buffer for 1 h, incubated with primary antibodies at 4 °C overnight, incubated with secondary antibodies at room temperature for 1 h, incubated with ABC reagent at room temperature for 45 min, incubated with activated DAB solution and mounted with Hematoxylin reagent. IHC signals were observed with the Keyence BZ-X700 microscope.

## Purification of GST-STAT6 proteins

The recombinant GST-STAT6-truncated proteins (aa 633–801) were purified from *E. coli* BL21 with isopropyl β-D-1-thiogalactopyranoside (IPTG) induction for 16 h at 16 °C. The bacteria pellets were resuspended in lysis buffer (0.5 mM EDTA, protease inhibitor cocktail in PBS) for sonication. Following centrifugation for 30 min at 16, 000×$g$ to get rid of bacteria debris, pre-washed GST-sepharose 4B beads slurry (GE Healthcare) were added to the supernatant and then incubated for 4 h at 4 °C. Beads were washed three times with ice-cold PBS at 1000×$g$. Proteins were eluted by elution buffer (20 mM reduced glutathione in 50 mM Tris-HCl, pH 8.0) and subjected to SDS-PAGE.

## JAK1 in vitro kinase assays

JAK1 in vitro kinase assays were adapted from a previous protocol. Briefly, 2 μg of purified GST-STAT6-truncated form from *E. coli* BL21 strain were incubated with immunoprecipitated Flag-JAK1-KM (WT, S571A, S571D, and K908A) by Flag-M2 agarose beads (Sigma) from HEK293T cells in the presence of 100 μM cold ATP in the kinase reaction buffer (New England Biolabs, B6022) for 30 min at 37 °C. The reaction was stopped by the addition of SDS-containing buffer and resolved by SDS-PAGE for western blotting analysis.

## Mouse xenograft assays

All mouse work has been reviewed and approved by the UNC Institutional Animal Care and Use Committee under IACUC#22-056. All animals were housed in the UNC GMB (Genetic Medicine Building) room UB31 with regular room temperature, dark/light cycle and humidity. UNC animal facility technicians/veterinarians regularly monitor animal health and maintain the housing environment in this facility. Mouse xenograft assays were performed as described previously[16]. Briefly, for a combination of irinotecan and UNC2025, $1 \times 10^7$ MHH-ES-1 cells were injected into the flank of 20 female nude mice (Jackson Laboratory, strain#: 002019, NU/J, 4 weeks old), and tumor-bearing mice were randomly divided into four groups prior to treatments. Irinotecan and UNC2025 were dissolved in normal saline. when tumors became visible after 5 days, mice were weekly treated with 10 mg/kg irinotecan given by intraperitoneal injection or every 3 days treated with 10 mg/kg UNC2025 given by intraperitoneal injection or both drugs. Tumor size was measured every three days with a digital caliper, and the tumor volume was determined with the formula: $L \times W^2 \times 0.5$, where L is the longest diameter and W is the shortest diameter. After 28 days, mice were sacrificed, and tumors were dissected and weighed. For a combination of irinotecan and filgotinib, $1 \times 10^7$ MHH-ES-1 cells were injected into the flank of 20 female nude mice (Jackson Laboratory, strain#: 002019, NU/J, 4 weeks old) and tumor-bearing mice were randomly divided into four groups. Filgotinib was dissolved in 4% DMSO + 30% PEG 300 + ddH$_2$O. When tumors became visible after 5 days, mice were weekly treated with 10 mg/kg irinotecan or 10 mg/kg filgotinib given by intraperitoneal injection or both drugs. Tumor size

was measured twice a week with a digital caliper, and the tumor volume was determined with the formula: $L \times W^2 \times 0.5$, where L is the longest diameter and W is the shortest diameter. After 25 days, mice were sacrificed, and tumors were dissected and weighed. For combination of irinotecan and MRX-2843, $1 \times 10^7$ MHH-ES-1 cells were injected into the flank of 40 female nude mice (Jackson Laboratory, strain#: 002019, NU/J, 4 weeks old) and tumor-bearing mice were randomly divided into eight groups prior to treatment. MRX-2843 were dissolved in normal saline. When tumors became visible after 8 days, mice were weekly treated with 10 mg/kg irinotecan given by intraperitoneal injection or twice a week treated with 10 mg/kg MRX-2843 given by intraperitoneal injection or both drugs, or twice a week treated with 10 mg/kg MRX-2843 given by oral gavage or both drugs, or twice a week treated with 30 mg/kg MRX-2843 given by oral gavage or both drugs. Tumor size was measured every 3 days with a digital caliper, and the tumor volume was determined with the formula: $L \times W^2 \times 0.5$, where L is the longest diameter and W is the shortest diameter. Mice with obvious tumor ulceration or tumor volume greater than 1800 mm$^3$ were removed from the study. The maximal tumor size/burden was not exceeded.

For Ewing sarcoma PDX xenograft assays, bulk NCH-EWS-1 PDX tumor stored in liquid nitrogen was dissociated into a single cell suspension. Briefly, the bulk PDX tumor was transferred into a 10 cm cell culture dish with sterile forceps in a laminar flow hood and cut into smaller pieces by a sterile scalpel. The tumor pieces were minced using a sterile razor blade. About 10 mL of 1x dissociation solution (8.9 mL of Media 199, 1 mL 10x collagenase/hyaluronidase (Stem Cell), 100 μL 100x DNase1 (Worthington)) was added to each of ~2 K mm$^3$ tumor. Gently agitate the dish to mix the solution with tumor slices and place the dish in a $CO_2$ incubator at 30 °C with 5% $CO_2$ for 15 min. Gently pipet the mixture up and down with serological pipettes and place the dish back in the $CO_2$ incubator. After at least three hours with ~12 repeats, check the dissociation status under a light microscope. Once the homogenous solution is obtained, pass the mixture through a 70 μm cell strainer and wash the strainer with 10 mL IMDM + 10% FBS. Centrifuge the collected suspension at 1000 rpm for 3 min. Aspirate the media, add 1 mL ice-cold ACK lysing buffer, and incubate for 3 min on ice. Cells were washed with sterile PBS and cell pellets were obtained by centrifugation. Cells were then resuspended in sterile PBS for cell counting (Bio-Rad TC20). 1x10e7 isolated PDX cells used for each xenograft injection mixed 1:1 with Matrigel into 5-week-old female nude mice (Jackson Lab 002019). Fresh NCH-EWS-1 PDX tumors were harvested and single PDX cells were isolated as described above. This round of PDX tumor amplification enabled enough PDX cells for injections into nude mice for different treatment groups. Mice xenografted with PDX cells were randomly divided into four groups. MRX-2843 were dissolved in normal saline. When tumors became visible after 4 days, mice were weekly treated with 10 mg/kg irinotecan given by intraperitoneal injection or twice a week treated with 10 mg/kg MRX-2843 given by intraperitoneal injection or both drugs. Tumor size was measured every 3 days with a digital caliper, and the tumor volume was determined with the formula: $L \times W^2 \times 0.5$, where L is the longest diameter and W is the shortest diameter. After 21 days, mice were sacrificed, and tumors were dissected and weighed.

## Cell viability assays

About $5 \times 10^3$ cells from indicated Ewing sarcoma cell lines were seeded into each well of 96-well plates. For combination treatment with SN-38 or etoposide and the indicated inhibitors, cell viability was measured 2 days after combination treatment using the CellTiter-Glo® Luminescent Cell Viability Assay (Promega, G7572) according to the manufacturer's instructions. For SN-38 or etoposide sensitivity assay, $5 \times 10^3$ cells from indicated Ewing sarcoma cell lines were seeded into each well of 96-well plates. Cell viability was measured 2 or 3 days after combination treatment using the CellTiter-Glo® Luminescent Cell

Viability Assay (Promega, G7572) following to the manufacturer's instructions. Three independent experiments were performed to generate error bars.

## Statistical analyses

Statistical analyses were performed using the SPSS 11.5 Statistical Software. $p \leq 0.05$ was considered statistically significant. The results are shown as means ± SD from at least two or three independent experiments as indicated in figure legends. Differences between control and experimental conditions were evaluated by one-way ANOVA. the synergy for a given combination is determined by the Chou–Talalay method based on the median-effect equation for combination index (CI) determination[29]. No statistical method was used to predetermine the sample size. No data were excluded from the analyses. The Investigators were not blinded to allocation during experiments and outcome assessment.

## Reporting summary

Further information on research design is available in the Nature Portfolio Reporting Summary linked to this article.

## Data availability

Data underlying the findings of this study are available in the Article, Supplementary and Source Data files. Source data are provided with this paper.

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

## Acknowledgements

We thank Liu lab members for their critical reading of the manuscript and helpful discussions. We sincerely thank Dr. Peter J. Houghton for sharing various Ewing sarcoma PDX tumors. We also sincerely thank Dr. Cote Jean-Francois for sharing various GAS6 constructs. We thank Dr. Siyuan Su for their help with the irinotecan-treated tumor samples. This work was supported by the Andrew McDonough B+ Foundation Childhood Cancer Research Grant (P.L.), Gabrielle's Angel Foundation Medical Research Award (P.L.), and the University of North Carolina at Chapel Hill University Cancer Research Fund (P.L.).

## Author contributions

L.Y.: Conceptualization, data curation, formal analysis, investigation, writing–original draft, writing–review and editing. Y.D.: formal analysis, investigation, writing–review and editing. X.W.: Resources. C.S.: Data curation. I.J.D.: Resources, writing–review and editing. H.S.E.: Resources, conceptualization, writing–review and editing. P.L.: Conceptualization, resources, supervision, funding acquisition, writing–review and editing.

## Competing interests

H.S.E. is a founder of Meryx (a UNC start-up) that is developing small molecule inhibitors for MERTK. H.S.E. and X.W. own stock in Meryx. The remaining authors declare no competing interests.
