## [Peer Review File · Nature Communications]

Reviewers' Comments:

Reviewer #1:

Remarks to the Author:

The manuscript by Yu et al. describes a study delving into the molecular mechanisms of chemoresistance of Ewing sarcoma tumours. Performing experiments on in vitro Ewing sarcoma cell cultures, the authors found that Akt and ERK signalling were major signalling mediators of resistance to a variety of chemotherapeutic agents. Also, the TAM kinases were found to be major activators of these kinase pathways in response to chemotherapeutic agents. Furthermore, the JAK-STAT6 signal axis was observed to lie upstream of TAM and GAS6 ligand upregulation in tumour cells, indicating autocrine/paracrine regulation. Genetic and pharmacological ablation of TAM kinases sensitised Ewing sarcoma cells and tumours to chemotherapy in vitro and in vivo. This also extended to a TAM small molecule inhibitor that is currently undergoing clinical trials, which showed synergism with chemotherapy to result in Ewing sarcoma tumour growth suppression in vivo. Thus, the study's findings have elucidated potential new selective molecular targets for therapeutic inhibition in Ewing sarcoma as a means to combat the development of chemoresistance.

The comprehensive study of chemoresistance mediators in Ewing sarcoma tumours is important and valuable, and the study has yielded many results. However, in order for the study's presented data to be more concrete and convincing, various aspects of the manuscript should be considered for improvement or refinement. Specific comments follow below.

1. Western blots. The manuscript is heavily dependent on W blot results. However, there is no indication throughout the manuscript as to how many times W blots were performed for each type of experiment. It is therefore important to state the number of experimental repeats performed. Ideally, densitometric quantification of W blots, from multiple repeats, would enable a clearer determination of the results as presented and interpreted. Below, examples are provided where the results displayed do not match the statements made about them.
2. Methods. Please state whether the cell lines used were authenticated and tested to be mycoplasma free.
3. Fig. 2 – It is not ideal to display separate W blots alongside each other, but with the 'control' (DMSO) sample lane only being present in one of them.
4. Fig. 2 – I would recommend that the growth curve results for cells subjected to sgMERTK-2 and agTYRO3-1 be left out of these results, as these specific treatments have not first been shown to affect the expression of the respective RTK. This could be done in a revision.
5. Page 7, line 164, Fig. 2 – the W blots do not show that MERTK (d) and AXL (e) depletion reduce pERK signals, as it is stated. It is clear with pAkt, however not so with pERK. In order for the results to be unequivocal, quantification analysis as suggested above would address this, especially with the total form of the protein being important as a comparator.
6. Page 10, line 243, Fig. 4 – the W blots do not show that GAS6 depletion "blocked" ERK activation by SN-38 (n) or etoposide (o). Rather, at most the W blots indicate that the effect is a reduced activation. Although the band signals are generally weaker, there are nevertheless clear activating effects observed with the agents under all conditions. Quantification would determine this more clearly, especially with normalisation against the total proteins.
7. Page 10, line 245, Fig. 4o – the W blots do not show that GAS6 depletion "blocked" ERK activation. Rather, the W blot indicates that the effect is a reduced activation at most. Quantification would determine this more clearly.
8. Page 8, line 182, Suppl Fig. 3d – it can also be added to the statement that IGF1-R depletion blocked SN-38 induced pAkt activation, as it did with pERK.
9. Page 9, line 206 – please justify the statement that 10mg/kg IP of UNC2025A is deemed a "low dose".

10. Page 10, line 227 – the statement that PROS was “not generally” regulated by the agents tested is perhaps not very helpful. It would be better to instead state it was regulated not as much as GAS6.

11. Page 11, line 250, please qualify the statement that the GAS6 in conditioned medium effect was “potent” in signalling activation. It would be useful to know the concentration of GAS6 in the medium.

12. Page 11, line 251, Fig 4u - please state in the Fig legend at least what concentration SN-38 was applied in this experiment.

13. Page 11, line 268, Fig 5f – there seems to be an error here – the results and legend show that the test agent is SN-38, not etoposide, as stated in the text. Please clarify.

14. Page 12, line 270, Fig 5g – this experiment would be more helpful and insightful if the cells were first pre-treated with sgSTAT6 prior to exposure to GAS6 conditioned medium.

15. Page 12, line 276, Fig 5g – the term “indispensable” is perhaps too exaggerated based on the results. I suggest “is a regulator of”.

16. Page 12, Fig 6c – The IF Fig is not convincing for showing clear differences in nuclear translocation across different treatments. I suggest both higher magnification images be displayed, as well as quantification done of the cytoplasmic:nuclear fluorescence ratio measures.

17. Fig. 6f-k – These experiments should include a lane that shows the results in cells treated with the different agents alone. They may have effects of baseline signal activation levels which should not be ignored, especially as W blotting captures relative signal levels rather than absolute.

Reviewer #2:

Remarks to the Author:

In this manuscript by Yu et al., the authors hypothesized that Ewing sarcoma cells acutely adapt to chemotherapy by altering specific signaling pathways prior to genomic changes and investigate the role of the JAK1-STAT6-GAS6-TAMK signaling network and its role in Ewing sarcoma chemosensitivity. The authors establish a chemotherapy-driven intrinsic mechanism of therapeutic resistance through the activation of JAK-STAT signaling, which then transcriptionally activates GAS6. GAS6 is then secreted and has an autocrine/paracrine effect on the tumors cells to activate the AKT and ERK pathways and direct chemoresistance. The mechanism and connections between the intrinsic activation to secretion of TAM activating ligand is well described as well as the step wise activation of the individual components of the signaling cascade. Also, they show that targeting components of this pathway have additive/synergistic effects in vivo. However, there are some concerns regarding the conceptual as well as technical aspects of the manuscript that need to be addressed.

Conceptual issues:

1. Patients treated for Ewing sarcoma see a combination of chemotherapy, which includes simultaneous vincristine, doxorubicin and Cytoxan which is then alternated with Ifosfamide and Etoposide for upfront therapy. For relapse therapy it is unusual for just a single agent to be administered as well. A typical relapsed therapy includes vincristine, irinotecan and temozolomide (VIT), and both Irinotecan and TMZ are given on days 1-5. So how will tumor cells react to combo that simulates actual therapy

So while the authors investigate the effects of single agent therapy, the mechanisms of resistance are most likely different upon combination therapy.

2. Also, for figure 1a-c results they use 1 cell line and 24h exposure to SN38 to identify their adoptive mechanisms of resistance, when at this time period it is difficult to conclude that these are actually mechanisms of resistance. There was no real proof that these are mechanisms associated with inhibiting apoptosis/necrosis in the models. Ewing sarcoma is known to be chemo

sensitive and presuming that these treatments at 24 hours are leading to chemoresistance is not well justified.

Technical comments:

Figure 1:

Explain the rationale for the concentrations used in Figure 1 (some seem quite high) as well as the timepoint for analysis at 24hrs. What happens if assess phospho-proteins at 12 hours or 48 hours? Would also be appropriate to measure timeline of drug treatment in the text.

What % of cells undergoing apoptosis at this timepoint?

Did the authors assess any combinations tried to see if effects p-status of pathways and try to recapitulate the patient treatment regimen

Were any upstream kinases, such as AXL, MET on the phosphoprofile? How come activation of these signaling components not noted? Please comment/address.

Fig 1m: Status of p-STAT6 and Gas6 expression in A673 xenografts?

Show tumor response? This still does not prove these are associated with driving resistance. Also, the authors should assess with multiple models and chemotherapies...not just irinotecan. For Fig 1m, they are biased towards looking at p-ERK and p-AKT. Should do phosphoarray analysis on these tumors? Other pathways that might be contributing to the "chemoresistance", which again needs to be more convincingly demonstrated.

Major issue (goes back to the conceptual point noted above) is that the authors use Fig 1 to claim chemoresistance without really proving that the cells are chemoresistant.

Fig 2d-f: Tried only SN-38 on MERTK/AXL depleted cells. They should assess response to other chemotherapies (noted in Figure 1) and additional cell line models.

What is the baseline signaling profile for these KO cells?

Suppl Fig 3 data re: IGFR-1--not adding much. Can't conclude that the lack of blocking ERK activation is the reason why its not overcoming resistance.

Fig. 3:

Better description of synergy plots...

Any toxicities with combo therapy.

Other models tested? PDXs?

Show that GAS6 is a direct target of STAT6? Or an indirect mechanism? Need to further prove or discuss.

Fig 3i: Check tumors treated with just irinotecan only for evidence of activation of these pathways/mechanisms of action..do you see evidence of the STAT6-GAS6-TAMK activation

Compared to Fig 1m (where not clear of the tumor size or response to Irinotecan) the investigator should do analysis of tumors post-treatment---assess effects on small molecule targets?

Any evidence of these being present in relapsed patient tumors?

Fig 5: Seem like very high doses of SN-38 needed for effects on cell viability...

see the effects of phosphorylation at low [], but not on viability when KO STAT6

What are the effects of phosphorylation at these higher doses of chemo? IC50s are 10fold higher than used for WBs.

At these higher concentrations (10-300uM) what other signaling pathways are being altered?

This needs to be discussed as well.

Fig 8. In vivo studies would benefit from assessing the effects on more than 1 cell line model.

Reviewer #3:

Remarks to the Author:

Yu and colleagues present an interesting study on signaling pathways that contribute to chemoresistance in Ewing sarcoma and provide data that nominate potential effective combination strategies. The authors provide evidence that chemotherapy induces phosphorylation/activation of multiple signaling proteins, notably AKT-T308, in Ewing sarcoma cells. The authors suggest that this chemo-induced signaling is dependent on TAM receptor activation via upregulation of Gas6 (presumably autocrine activation). The authors show that Gas6 induction is STAT6-mediated and that STAT6 is activated by JAK1, which itself is stimulated as a result of chemotherapy. Therapeutic implications are highlighted through xenograft studies where inhibition of JAK1 or TAM receptors in combination with chemotherapy potentially reduces Ewing xenograft growth.

There are multiple noteworthy results. 1. Demonstration of a plausible pathway that induces TAM (Axl/MerTK) activation by chemotherapy; 2. Use of multiple Ewing cell lines increases rigor in the study; 3. Loss/Gain of function and pharmacologic inhibition of the TAM pathway results in consistent phenotypes in multiple cell lines, again increasing rigor; 3. Activation of JAK1 by chemo is a novel aspect of the study and the authors provide evidence to support this claim.

The work is original. There is limited published data on the contribution of Axl, MerTK, JAK1 and STAT6 individually in Ewing sarcoma. The concept that combination therapy is needed to improve outcomes in Ewing sarcoma is not novel. However, the strategy of looking for chemo-activated pathways in tumor cells and then targeting those pathways as a route to an effective combo is novel.

The methods are adequately described (except where noted below) and the study encompasses a wide breadth of techniques, results of which in general support the overall conclusions of the study.

Major Comments:

1. In general additional detail should be added to the figure legends and/or methods section where appropriate. For example, Figure 1B, the vendor and cat # for the membrane used to probe phospho-proteins should be provided and the conditions used to stimulate cells before lysate production should be clarified (e.g., how long were cells stimulated with SN-38). A key detail that should be provided in the figure legends is time frame of stimulation or assay.

2. Was AKT-pS473 included in the membrane shown in Fig 1B? The authors only show data with AKT-pT308, why? If AKT-s473 is not phosphorylated is the AKT pathway relevant to chemoresistance?

3. The activation of JAK1 by chemotherapy needs to be clarified. The text presents the concept that DNA damage response kinases (ATM, ATR, DNAPK) can phosphorylate JAK1 at S571 and this is repeated in the discussion. However, the authors do not mention that a SQ sequence is indicative of an ATM/ATR substrate site. It will be confusing for readers to just jump to JAK1-pSQ terminology. As written there are sections of the results that read as if JAK1 activation occurs in the absence of DNA damage response. Please clarify.

4. Does activation of JAK1 by chemo occur in the nucleus or cytoplasm?

5. The in vivo efficacy studies are promising. I strongly suggest/recommend repeating in vivo studies testing the best combo therapy vs chemo alone and control (limited groups) in mice with established tumors (>250 mm³) to drive higher impact.

Minor:

1. In Fig 2 the authors show that KD of AXL blocks chemo-induced AKT-pT308 but cabozantinib, a reasonably potent inhibitor of AXL, does not. This is somewhat confusing and should be discussed.
2. I recommend showing a cell viability curve for at least a few of the combination cell viability studies displayed in Fig 3a-h. Just as examples, these could be in the supplement. The heat maps are not intuitive.
3. I would avoid the use of JAK1-KD terminology. 'KD' is commonly used for knockdown or kinase dead, which can easily lead to reader confusion.
4. Fig 6c is unconvincing as displayed. Either replace, move to supplement or delete. The cell fractionation is sufficient to demonstrate nuclear localization.

Suggested edits:

Line 237 – change 'BCA' to 'TCA'

Line 260 – change 'macrophage' to 'macrophages'

Reviewer #4:

Remarks to the Author:

This manuscript reports very interesting studies on the understanding of the molecular basis of Ewing sarcoma chemoresistance. The authors provide strong data to suggest that chemotherapy induces JAK1 activation via DNA damaging kinases-mediated JAK1-SQ phosphorylation, which leads to the subsequent activation of STAT6, resulting in the transcriptional activation of GAS6. The secretion of GAS6 protein induces TAM kinase activation and then the increased activation of Akt and ERK in Ewing sarcoma cells. The authors propose that this newly identified JAK1/STAT6/GAS6/TAM/Akt-ERK pathway contributes to Ewing sarcoma chemoresistance. In support of their hypothesis, JAK1 inhibitor filgotinib or TAM kinase inhibitor UNC2025A sensitized Ewing sarcoma cells and tumors to chemotherapy in vitro as well as in animal tumor models. In addition, the TAM kinase inhibitor MRX-2843 currently in clinical trials to treat AML and advanced solid tumors showed synergistic effects with chemotherapy to inhibit Ewing sarcoma tumor growth. These results have potential translational value to overcome Ewing sarcoma chemoresistance. There are a few issues that the authors should address.

- 1) Several previous studies have reported the involvement of STAT3 in the development of chemoresistance, and GAS6 is known to induce the activation of STAT3. The authors should provide experimental data to address if STAT3 plays any role in Ewing sarcoma chemoresistance.
- 2) Have the authors tested if JAK1 becomes tyrosine phosphorylated/activated by chemo drugs in Ewing sarcoma cells?
- 3) Can the authors show the direct binding of STAT6 to the promoter of GAS6 gene by ChIP assays?
- 4) Serine phosphorylation of STATs is important for STAT-mediated gene activation. Several kinases including ERKs are involved in STAT serine phosphorylation. Since chemotherapy was shown to induce ERK activation in Ewing sarcoma cells, does the chemotherapy-induced ERK activation have any effect on STAT serine phosphorylation, if so, does this phenomenon contribute to the chemoresistance in Ewing sarcoma?

Point-to-Point Responses to the Reviewers' Critiques (NCOMMS-23-43600)

We deeply appreciate the constructive suggestions provided by the editor and four reviewers during the initial review of our manuscript. The comments and suggestions have been very helpful in guiding us to further improve our study. Over the past 3 months, we have obtained additional experimental evidence to further support our conclusions and extensively improved our manuscript. We hope that the editor and the reviewers concur with us that we have fully addressed all the reviewers' concerns and substantially strengthened our paper. Therefore, we believe that the revised manuscript is now suitable for publication in *Nature Communications*.

Reviewer #1

The manuscript by Yu et al. describes a study delving into the molecular mechanisms of chemoresistance of Ewing sarcoma tumours. Performing experiments on in vitro Ewing sarcoma cell cultures, the authors found that Akt and ERK signalling were major signalling mediators of resistance to a variety of chemotherapeutic agents. Also, the TAM kinases were found to be major activators of these kinase pathways in response to chemotherapeutic agents. Furthermore, the JAK-STAT6 signal axis was observed to lie upstream of TAM and GAS6 ligand upregulation in tumour cells, indicating autocrine/paracrine regulation. Genetic and pharmacological ablation of TAM kinases sensitised Ewing sarcoma cells and tumours to chemotherapy in vitro and in vivo. This also extended to a TAM small molecule inhibitor that is currently undergoing clinical trials, which showed synergism with chemotherapy to result in Ewing sarcoma tumour growth suppression in vivo. Thus, the study's findings have elucidated potential new selective molecular targets for therapeutic inhibition in Ewing sarcoma as a means to combat the development of chemoresistance.

The comprehensive study of chemoresistance mediators in Ewing sarcoma tumours is important and valuable, and the study has yielded many results. However, in order for the study's presented data to be more concrete and convincing, various aspects of the manuscript should be considered for improvement or refinement. Specific comments follow below.

Response: We thank the reviewer for acknowledging the importance and therapeutic potential for our study. We also thank the reviewer for raising these great suggestions to further improve our manuscript. Please find our detailed point-to-point responses to each raised question below.

1. Western blots. The manuscript is heavily dependent on W blot results. However, there is no indication throughout the manuscript as to how many times W blots were performed for each type of experiment. It is therefore important to state the number of experimental repeats performed. Ideally, densitometric quantification of W blots, from multiple repeats, would enable a clearer determination of the results as presented and interpreted. Below, examples are provided where the results displayed do not match the statements made about them.

Response: We thank the reviewer for raising this important suggestion. We have been always working hard to ensure reproducibility of our data. In the revised manuscript, we have included statements for how many times each presented experiments were performed in figure legends. In addition, as suggested below, we have also provided quantifications for western blot analyses in the revised Fig. 2d-2f, Fig. 4n, 4o and revised Supplementary Fig. 3a-3l.

2. Methods. Please state whether the cell lines used were authenticated and tested to be mycoplasma free.

Response: We thank the reviewer for raising this question. We have included this statement in the revised method section.

3. Fig. 2 – It is not ideal to display separate W blots alongside each other, but with the 'control' (DMSO) sample lane only being present in one of them.

Response: We thank the reviewer for raising this question. We have repeated part of this experiment and included

the control as suggested (revised Fig. 2c).

4. Fig. 2 – I would recommend that the growth curve results for cells subjected to sgMERTK-2 and agTYRO3-1 be left out of these results, as these specific treatments have not first been shown to affect the expression of the respective RTK. This could be done in a revision.

Response: We thank the reviewer for this suggestion. In the revised manuscript, we have included newly obtained experimental data showing sgMERTK-2 cells behaved similarly as sgMERTK-1 cells (revised Fig. 2d, revised Supplementary Fig. 3a, 3d, 3g, 3j).

5. Page 7, line 164, Fig. 2 – the W blots do not show that MERTK (d) and AXL (e) depletion reduce pERK signals, as it is stated. It is clear with pAkt, however not so with pERK. In order for the results to be unequivocal, quantification analysis as suggested above would address this, especially with the total form of the protein being important as a comparator.

Response: We thank the reviewer for raising this question. Following the reviewer's suggestions, we have repeated these assays, and also included quantifications for key western blots as in the revised Fig. 2d-2f and revised Supplementary Fig. 3a-3i.

6. Page 10, line 243, Fig. 4n – the W blots do not show that GAS6 depletion “blocked” ERK activation by SN-38 (n) or etoposide (o). Rather, at most the W blots indicate that the effect is a reduced activation. Although the band signals are generally weaker, there are nevertheless clear activating effects observed with the agents under all conditions. Quantification would determine this more clearly, especially with normalisation against the total proteins.

Response: We thank the reviewer for raising this question and fully agree that quantifications improve the data interpretations. Following the reviewer's suggestions, we have also included quantifications for key western blots as in the revised Fig. 4n, 4o to clearly indicated the changes of ERK-p42/p44 signaling. In addition, we have replaced “blocked” with “alleviated” to tone down this effect in the revised manuscript.

7. Page 10, line 245, Fig. 4o – the W blots do not show that GAS6 depletion “blocked” ERK activation. Rather, the W blot indicates that the effect is a reduced activation at most. Quantification would determine this more clearly.

Response: We thank the reviewer for raising this question and fully agree that quantifications improve the data interpretations. Following the reviewer's suggestions, we have also included quantifications for key western blots as in the revised Fig. 4n, 4o to clearly indicated the changes of ERK-p42/p44 signaling. In addition, we have replaced “blocked” with “alleviated” to tone down this effect in the revised manuscript.

8. Page 8, line 182, Suppl Fig. 3d – it can also be added to the statement that IGF1-R depletion blocked SN-38 induced pAkt activation, as it did with pERK.

Response: We thank the reviewer for this suggestion and have revised the statement as suggested in the revised manuscript.

9. Page 9, line 206 – please justify the statement that 10mg/kg IP of UNC2025A is deemed a “low dose”.

Response: We thank the reviewer for raising this critical question. In previous studies, UNC2025A was used in animals at doses of 50 mg/kg or 75 mg/kg (DeRyckere et al. Clinical Cancer Research. 2017. 23(6): 1481-1492). We have included this information in the revised manuscript.

10. Page 10, line 227 – the statement that PROS was “not generally” regulated by the agents tested is perhaps not very helpful. It would be better to instead state it was regulated not as much as GAS6.

Response: We thank the reviewer for this advice. In the revised manuscript, we have revised this statement as

suggested.

11. Page 11, line 250, please qualify the statement that the GAS6 in conditioned medium effect was “potent” in signalling activation. It would be useful to know the concentration of GAS6 in the medium.

Response: We thank the reviewer for raising this question. As suggested, we removed the statement of “potent” to tone down this statement. In addition, we also immunoprecipitated secreted GAS6 proteins from 1×10^7 MHH-ES-1 cells treated with 1 μ M SN-38 for 24 hrs and compared the amount of GAS6 proteins to bacterially purified proteins with a defined concentration. From the comparison and calculation, roughly 1.5 μ g GAS6 proteins were obtained from 1×10^7 MHH-ES-1 cells.

12. Page 11, line 251, Fig 4u - please state in the Fig legend at least what concentration SN-38 was applied in this experiment.

Response: We apologize for missing this information. We included the SN-28 concentration (1 μ M) in both revised Fig. 4u and other figure legends.

13. Page 11, line 268, Fig 5f – there seems to be an error here – the results and legend show that the test agent is SN-38, not etoposide, as stated in the text. Please clarify.

Response: We apologize for this mistake and have corrected it as “SN-38” instead of “etoposide” in the revised manuscript. In addition, we have also thoroughly proof-read the manuscript to further eliminate similar typos.

14. Page 12, line 270, Fig 5g – this experiment would be more helpful and insightful if the cells were first pre-treated with sgSTAT6 prior to exposure to GAS6 conditioned medium.

Response: We thank the reviewer for raising this question. In this experiment, we generated the STAT6-depleted MHH-ES-1 cells first by sgRNA-mediated lentiviral infection followed by puromycin selection to eliminate non-infected cells. Resulting cells were cultured in normal condition or GAS6 conditioned media as indicated in the figure. We have included more details in the figure legend to avoid future confusions.

15. Page 12, line 276, Fig 5g – the term “indispensable” is perhaps too exaggerated based on the results. I suggest “is a regulator of”.

Response: We fully agree with the reviewer and have replaced “indispensable” with “is a regulator of” in the revised manuscript.

16. Page 12, Fig 6c – The IF Fig is not convincing for showing clear differences in nuclear translocation across different treatments. I suggest both higher magnification images be displayed, as well as quantification done of the cytoplasmic:nuclear fluorescence ratio measures.

Response: We thank the reviewer for this suggestion and apologize for the low-resolution images provided in the original submission. Following the reviewer’s suggestions, we have repeated this experiment and obtained higher magnification images to clearly indicate that upon SN-38 treatment, STAT6 translocates into nucleus with a quantification of these nuclear translocation events (revised Fig. 6c).

17. Fig. 6f-k – These experiments should include a lane that shows the results in cells treated with the different agents alone. They may have effects of baseline signal activation levels which should not be ignored, especially as W blotting captures relative signal levels rather than absolute.

Fig 6i-6k: 4 lanes including inhibitor treatment alone.

Response: We thank the reviewer for raising this great question and fully agree that we should also examine effects of inhibitor treatment alone. We have repeated these experiments and included inhibitor alone conditions. As shown in the revised Fig. 6f-6k, inhibitor alone didn’t significantly affect Akt or ERK activation.

Reviewer #2

In this manuscript by Yu et al., the authors hypothesized that Ewing sarcoma cells acutely adapt to chemotherapy by altering specific signaling pathways prior to genomic changes and investigate the role of the JAK1-STAT6-GAS6-TAMK signaling network and its role in Ewing sarcoma chemosensitivity. The authors establish a chemotherapy-driven intrinsic mechanism of therapeutic resistance through the activation of JAK-STAT signaling, which then transcriptionally activates GAS6. GAS6 is then secreted and has an autocrine/paracrine effect on the tumors cells to activate the AKT and ERK pathways and direct chemoresistance. The mechanism and connections between the intrinsic activation to secretion of TAM activating ligand is well described as well as the step wise activation of the individual components of the signaling cascade. Also, they show that targeting components of this pathway have additive/synergistic effects in vivo. However, there are some concerns regarding the conceptual as well as technical aspects of the manuscript that need to be addressed.

Response: We thank the reviewer for acknowledging the importance and therapeutic potential for our study. We also thank the reviewer for raising these great suggestions to further improve our manuscript. Please find our detailed point-to-point responses to each raised question below.

Conceptual issues:

1. Patients treated for Ewing sarcoma see a combination of chemotherapy, which includes simultaneous vincristine, doxorubicin and Cytoxan which is then alternated with Ifosfamide and Etoposide for upfront therapy. For relapse therapy it is unusual for just a single agent to be administered as well. A typical relapsed therapy includes vincristine, irinotecan and temozolomide (VIT), and both Irinotecan and TMZ are given on days 1-5. So how will tumor cells react to combo that simulates actual therapy. So while the authors investigate the effects of single agent therapy, the mechanisms of resistance are most likely different upon combination therapy.

Response: We thank the reviewer for raising this critical question and fully agree with the reviewer that initial therapy for localized Ewing sarcoma involves 5 drugs used in an alternating pattern, including doxorubicin and etoposide. However, in the context of recurrence, irinotecan (which is metabolized to the topoisomerase I inhibitor, SN-38) and temozolomide (TMZ) have been shown to be temporarily effective for a large fraction of patients (Casey et al. *Pediatr Blood Cancer*. 2009. 53: 1029-1034), and this combination has been used together with experimental agents (Federico et al., 2020. *Eur J Cancer*. **137**: 204-213; Wang et al. 2022. *BMC Cancer*. **22**: 349).

We fully agree with the reviewer that it is critical to test if combo-chemo behaves similar to single chemo in our signaling studies. We compared the combination of SN-38 with TMZ to each single treatment in three Ewing sarcoma cells at doses that neither of the treatment killed cells in a 24-hr period, and found that combination treatment further increased ERK-p42/p44 with minimal effects on Akt-pT308 compared with each single chemo-treatment in A673, MHH-ES-1 and TC-71 cells (revised Supplementary Fig. 1i-1k). In addition, isoflavone has also been used as an alternative upfront therapy (Zollner et al. *Journal of Clinical Medicine*. 2021. 10(8):1685). We further tested effects of isoflavone in regulating activation of these kinases. We observed that similar to other chemotherapeutics we tested, isoflavone also induced Akt-pT308 and ERK-p42/p44 in A673 (revised Supplementary Fig. 1l), MHH-ES-1 (revised Supplementary Fig. 1m) and TC-71 cells (revised Supplementary Fig. 1n). Furthermore, combination of isoflavone with SN-38 further promoted activation of both Akt and ERK (revised Supplementary Fig. 1o-1q). Together, these data suggest that chemotherapeutic agents (both alkylator and topoisomerase inhibitors) may generally activate Akt/ERK signaling in Ewing sarcoma cells.

2. Also, for figure 1a-c results they use 1 cell line and 24h exposure to SN38 to identify their adoptive mechanisms of resistance, when at this time period it is difficult to conclude that these are actually mechanisms of resistance. There was no real proof that these are mechanisms associated with inhibiting apoptosis/necrosis in the models. Ewing sarcoma is known to be chemo sensitive and presuming that these treatments at 24 hours are leading to chemoresistance is not well justified.

Response: We thank the reviewer for raising this excellent question. To answer this question, we have performed additional experiments as listed below.

1. We have obtained new experiment data showing that (1) 24-hr chemo-treatment has significantly triggered Ewing sarcoma cell apoptosis as evidenced by increased cleaved-PARP1 and cleaved-caspase-3 signals (revised Fig. 4b, 4c and revised Supplementary Fig. 6a-6e); (2) 24-hr chemo-treatment has significantly activated Akt and ERK kinases in Ewing sarcoma cells as evidenced by increased Akt-pT308 and ERK-p42/p44 signals (revised Fig. 1f-1i and revised supplementary Fig. 1a-1q); (3) inhibiting Akt, ERK or TAM kinases by MK2206, PD0325901, or UNC2025A, respectively, all led to enhanced cell apoptosis evidenced by increased cleaved-PARP1 and cleaved-caspase-3 signals (revised Supplementary Fig. 6f). These data suggest that increased Akt/ERK activation upon chemotherapy in Ewing sarcoma partially contributes to resistance to 24-hr chemo-treatment.
2. In addition, to further test if JAK/GAS6/TAM signaling plays roles in Ewing sarcoma chemoresistance, we have isolated SN-38 resistance single clones from MHH-ES-1 cells following a 2-month selection with $\frac{1}{2}$ IC50 SN-38 treatment. We found these resistant clones were indeed displayed significantly increased resistance to SN-38 treatment *in vitro* (Fig. R1a), and increased Akt-pT308 and ERK-p42/p44 were observed in these resistant clones (Fig. R1b). Importantly, these SN-38 resistant MHH-ES-1 single clones were more sensitive to filgotinib treatment (JAK1 inhibitor) (Fig. R1c). These data suggest that at least some SN-38 resistant MHH-ES-1 cells display increased Akt/ERK signaling and are sensitized by our identified JAK/GAS6/TAM inhibition.

Figure R1. Isolation and characterization of SN-38 resistant MHH-ES-1 single clones. (a) MTT cell viability analyses showing isolated SN-38 resistant SR1-SR4 clones are resistant to SN-38 treatment *in vitro*. (b) IB analyses of WCL from indicated MHH-ES-1 cells showing increased Akt and ERK activities are observed in SR cells. (c) MTT cell viability assays showing that MHH-ES-1 SR cells are sensitive to filgotinib treatments.

We are of the opinion that chemo-induced Akt and ERK activation serves as an early signaling event to protect cells from cell death, allowing cells to developing resistance such as long-lasting and stable genetic changes to confer to chemoresistance. We have also included more statements in the revised discussion section to emphasize this notion.

Technical comments:

3. Figure 1: Explain the rationale for the concentrations used in Figure 1 (some seem quite high) as well as the timepoint for analysis at 24hrs. What happens if assess phospho-proteins at 12 hours or 48 hours? Would also be appropriate to measure timeline of drug treatment in the text.

Response: We thank the reviewer for raising this question. From the literature, given it is difficult to determine the comparable doses of chemotherapeutics to clinical doses, different doses of these chemotherapeutic agents have been used in Ewing sarcoma cell line studies. For SN-38, 1 μ M concentration is commonly used in previous Ewing sarcoma studies in either Ewing sarcoma cell lines (Grohar et al. *Clinical Cancer Research*. 2013. 20(5): 1190-1203) or Ewing sarcoma PDX cells (Castillo-Ecija et al. *Journal of Controlled Release*. 2020. 324: 440-449). For etoposide, 10 μ M treatment has been used (Luo et al. *Oncogene*. 2009. 26(46): 4126-4132) as well as 5 μ M (Boehme et al. *International Journal of Oncology*. 2016. 49(5): 2135-2146). For doxorubicin, up to 10 μ M has been used (Luo et al. *Oncogene*. 2009. 26(46): 4126-4132), and for TMZ, up to 300 μ M has been used (Song et al. *Scientific Reports*. 2016. 6: 22762). We tried a series of doses of each chemotherapeutic agents and presented the data in the revised Fig. 1f-1i and revised Supplementary Fig. 1).

We fully agree with the reviewer that both doses and treatment periods for chemotherapeutics are critical

for their effects on cell signaling and cell viability. In the original Supplementary Fig. 1c and 1d, we have included data measuring shorter periods of SN-38 treatment on both A673 and MHH-ES-1 cells and found increased Akt/ERK signaling could be observed as early as 2 hrs post-SN-38 treatment. Following the reviewer's suggestion, we included newly obtained experimental data showing that SN-38 and TMZ increased Akt and ERK activities at 12, 24 and 48 hrs respectively (revised Supplementary Fig. 1e, 1f). Interestingly, SN-38 continued to increase pERK signals with treatment time, however, Akt-pT308 signals peaked at 24 hrs. This was one of the reasons for our analyses on effects of chemotherapeutics on signaling performed at 24 hr post-treatment.

4. What % of cells undergoing apoptosis at this timepoint?

Response: We thank the reviewer for this question. In our originally submitted manuscript, we have performed FACS analyses to measure % of apoptotic cells 24 hrs post SN-38 or etoposide treatment. The apoptotic cells were increased from 4.15% (Q3, early apoptosis) and 5.35% (Q2, late apoptosis) to 28.3% (Q3) and 27.3% (Q2) by SN-38, and to 23.9% (Q3) and 22.8% (Q2) by etoposide (revised Fig. 4e). In addition, in the revised manuscript, we also included newly obtained data showing that 24-hr SN-38 treatment induced significant apoptosis in multiple Ewing sarcoma cells including MHH-ES-1, A673 and TC-71 (revised Fig. 4b, 4c, Supplementary Fig. 6a-6e).

5. Did the authors assess any combinations tried to see if effects p-status of pathways and try to recapitulate the patient treatment regimen.

Response: We fully agree with the reviewer that it is critical to test if combo-chemo behaves similar to single chemo in our signaling studies. We compared the combination of SN-38 with TMZ to each single treatment in three Ewing sarcoma cells at doses that neither of the dose killed cells at the 24-hr period, and found that chemo-combination treatment further increased ERK-p42/p44 with minimal effects on Akt-pT308 compared with each single chemo-treatment in three Ewing sarcoma cells including A673, MHH-ES-1 and TC-71 (revised Supplementary Fig. 1i-1k). In addition, isoflavone has also been used as an alternative for upfront therapy (Zollner et al. *Journal of Clinical Medicine*. 2021. 10(8):1685). We further tested effects of isoflavone in regulating activation of these kinases. We observed that similar to other chemotherapeutics we tested, isoflavone also induced Akt-pT308 and ERK-p42/p44 in A673 (revised Supplementary Fig. 1l), MHH-ES-1 (revised Supplementary Fig. 1m) and TC-71 cells (revised Supplementary Fig. 1n). In addition, combination of isoflavone with SN-38 further promoted activation of both Akt and ERK (revised Supplementary Fig. 1o-1q). Together, these data suggest that chemotherapeutic agents (both alkylator and topoisomerase inhibitors) can generally activate Akt/ERK signaling in Ewing sarcoma cells.

6. Were any upstream kinases, such as AXL, MET on the phosphoprofile? How come activation of these signaling components not noted? Please comment/address.

Response: We thank the reviewer for raising this great question. We used the Proteome Profiler Human Phosphokinase Array Kit (Biotechne # ARY003C), which includes 37 kinases in total. This panel does not include AXL, MET, MERTK, nor TYRO3. That is why these kinases were not identified through this profiling. We have included a new method section to describe the procedure and reagents for this phospho-profiling in the revised manuscript.

7. Fig 1m: Status of p-STAT6 and Gas6 expression in A673 xenografts?

Response: We thank the reviewer for this great question. We have performed IHC analyses on isolated tumors from this experiment and observed increased ERK-p42/p44 and pSTAT6 signals in irinotecan treated tumors compared with vehicle control treated tumors (revised Supplementary Fig. 2e). We also tried GAS6 staining in this IHC analysis, but we tried various GAS6 antibodies and failed to find any signals. Since GAS6 is a secreted protein, we thought most secreted GAS6 proteins were lost during tumor fixation steps which yielded low signals. In addition, we also performed TMZ treatment on xenografted A673 tumors, and found from IHC analysis that increased ERK-p42/p44 and STAT6-pY641 were also increased in TMZ treated tumors (revised Supplementary

Fig. 2b).

8. Show tumor response? This still does not prove these are associated with driving resistance. Also, the authors should assess with multiple models and chemotherapies...not just irinotecan. For Fig 1m, they are biased towards looking at p-ERK and p-AKT. Should do phosphoarray analysis on these tumors? Other pathways that might be contributing to the “chemoresistance”, which again needs to be more convincingly demonstrated.

Response: We thank the reviewer for raising these questions. Following the reviewer’s suggestions, we have also performed A673 xenograft experiment treated with TMZ in addition to irinotecan. As shown in the revised Fig. 1m-1o, TMZ treatment reduced tumor growth in nude mice with minimal effects on animal weights (revised Supplementary Fig. 2a). Analyses of isolated tumors showed that compared with vehicle control, TMZ treatment also increased tumor Akt-pT308 and ERK-p42/p44 signals (revised Fig. 1p), which was further confirmed by tumor IHC analysis (revised Supplementary Fig. 2b). These newly included data suggest chemotherapeutic treatment by either irinotecan or TMZ caused activation of Akt and ERK in tumors.

We fully agree with the reviewer that although these chemotherapeutic agents induced activation of Akt/ERK in these tumors, it does not necessarily mean activated Akt/ERK signaling contributes to chemoresistance. Our goal in this study is to improve chemotherapy efficacy instead of overcoming chemoresistance. We are of the opinion that Ewing sarcoma tumors may acutely increase Akt/ERK signaling upon chemotherapy to provide a cell survival protection, a period allowing Ewing sarcoma tumors to develop stable resistance mechanisms such as genetic changes. Thus, if we shorten this protection window by inhibiting JAK/TAM/GAS6 signaling as we performed in this study, this would improve chemotherapy efficacy. We have also further clarified this statement in the discussion section in the revised manuscript.

In addition, to further test if JAK/GAS6/TAM signaling plays roles in Ewing sarcoma chemoresistance, we have isolated SN-38 resistance single clones from MHH-ES-1 cells following a 2-month selection with 1/2 IC50 SN-38 treatment. We found these resistant clones were indeed displayed significantly increased resistance to SN-38 treatment *in vitro* (Fig. R1a), and increased Akt-pT308 and ERK-p42/p44 were observed in these resistant clones (Fig. R1b). Importantly, these SN-38 resistant MHH-ES-1 single clones were more sensitive to filgotinib treatment (JAK1 inhibitor) (Fig. R1c). These data suggest that at least some SN-38 resistant MHH-ES-1 cells display increased Akt/ERK signaling and are sensitized by our identified JAK/GAS6/TAM inhibition.

Figure R1. Isolation and characterization of SN-38 resistant MHH-ES-1 single clones. (a) MTT cell viability analyses showing isolated SN-38 resistant SR1-SR4 clones are resistant to SN-38 treatment *in vitro*. (b) IB analyses of WCL from indicated MHH-ES-1 cells showing increased Akt and ERK activities are observed in SR cells. (c) MTT cell viability assays showing that MHH-ES-1 SR cells are sensitive to filgotinib treatments.

Moreover, we fully agree with the reviewer it is a great idea to perform phospho-profiling using xenografted tumors treated with chemotherapy than using Ewing sarcoma cell lines. We have tried this assay using tumor lysates previously but failed. One major technical reason is because the tumors from mice are with quite many blood or blood vesicles, which generates a quite high background noise signals for high-sensitive assays like phospho-profiling. The advantage for using cell lines is that the population of cells are more unified with less heterogeneity and without contaminations from blood. Another reason is that from tumor harvest to obtaining tumor lysates takes more steps and longer time than simply harvesting cultured Ewing sarcoma cells on cell culture dishes. This prolonged processing steps and time would in theory affect signaling dynamics. Thus,

we are of the opinion to perform the phospho-profiling using cell lines but further validate key signaling events in tumors.

9. Major issue (goes back to the conceptual point noted above) is that the authors use Fig 1 to claim chemoresistance without really proving that the cells are chemoresistant.

Response: We fully agree with the reviewer that although these chemotherapeutic agents induced activation of Akt/ERK in these tumors, it does not necessarily mean activated Akt/ERK signaling contributes to chemoresistance. Our goal in this study is to improve chemotherapy efficacy instead of overcoming chemoresistance. We are of the opinion that Ewing sarcoma tumors may increase Akt/ERK signaling upon chemotherapy to provide a cell survival protection, a period allowing Ewing sarcoma tumors to develop stable resistance mechanisms such as genetic changes. Thus, if we shorten this protection window by inhibiting JAK/TAM/GAS6 signaling as we proposed in this study would improve chemotherapy efficacy. We have also further clarified this statement in the discussion section in the revised manuscript.

In addition, as illustrated in answers to last question, to further test if JAK/Gas6/TAM signaling plays roles in Ewing sarcoma chemoresistance, we have isolated SN-38 resistance single clones from MHH-ES-1 cells following a 2-month selection with ½ IC50 SN-38 treatment. We found these resistant clones were indeed displayed significantly increased resistance to SN-38 treatment *in vitro* (Fig. R1a), and increased Akt-pT308 and ERK-p42/p44 were observed in these resistant clones (Fig. R1b). Importantly, these SN-38 resistant MHH-ES-1 single clones were more sensitive to filgotinib treatment (JAK1 inhibitor) (Fig. R1c). These data suggest that at least some SN-38 resistant MHH-ES-1 cells display increased Akt/ERK signaling and are sensitized by our identified JAK/GAS6/TAM inhibition.

10. Fig 2d-f: Tried only SN-38 on MERTK/AXL depleted cells. They should assess response to other chemotherapies (noted in Figure 1) and additional cell line models.

Response: We thank the reviewer for this suggestion. Following the reviewer's suggestion, we have also tested effects of other chemotherapeutic agents on MERTK/AXL/TYRO3 depleted MHH-ES-1 cells. As shown in the revised Supplementary Fig. 3a-3i, in MHH-ES-1 cells depleted of either endogenous MERTK, AXL or TYRO3, we further tested how different chemotherapy affects Akt-pT308 and ERK-p42/p44. Our newly obtained data suggest that in responding to either SN-38, doxorubicin, etoposide or TMZ, depletion of either MERTK or TYRO3 but not AXL reduced chemo-induced activation of Akt and ERK. In addition, we confirmed that in A673 cells depletion of MERTK or TYRO3 but not AXL blocked SN-38-induced Akt-pT308 and ERK-p42/p44 signals (revised Supplementary Fig. 3j-3l).

11. What is the baseline signaling profile for these KO cells?

Response: We thank the reviewer for this question. We have included baseline Akt and ERK signaling in these KO cells in the revised Fig. 2d-2f and revised Supplementary Fig. 3m, 3p, 3s. We found that MERTK depletion mildly reduced basal ERK activity, while depletion other TAM kinase members didn't affect basal activity of Akt nor ERK.

12. Suppl Fig 3 data re: IGFR-1--not adding much. Can't conclude that the lack of blocking ERK activation is the reason why its not overcoming resistance.

Response: We thank the reviewer for bringing up this question and we fully agree with the reviewer that lack of blocking ERK activation might not be the failure reason for anti-IGF1R therapy in sensitizing Ewing sarcoma to chemotherapy. As suggested, we have toned down this statement in the revised manuscript.

13. Fig. 3: Better description of synergy plots... Any toxicities with combo therapy. Other models tested? PDXs?

Response: We apologize for the brief description about the synergy plots. We have included more details for these plots in the revised figure legends. In addition, we have also included more detailed cell viability data

supporting these synergy plots in the revised Supplementary Fig. 5a.

In addition, we have further strengthened our conclusion by delaying the treatment till xenografted tumors became larger. In the revised Supplementary Fig. 9b-9e, we xenografted MHH-ES-1 cells to nude mice and didn't start the treatment until tumors are at least 250 mm³. Similar to starting treatment earlier (revised Fig. 8l-8o), we observed that even for large tumors that more faithfully mimic clinical settings, compared with single treatment by either irinotecan or MRX-2843, the combination significantly improved the treatment efficacy in suppressing tumor growth (revised Supplementary Fig. 10b-10d). Importantly, no toxicity was observed associated with these treatments, either for early treatments (revised Fig. 8o) or late treatments (revised Supplementary Fig. 10e).

We fully agree with the reviewer that PDX is a critical and more clinically related model to test the synergistic effects of our proposed combination therapy. Following the reviewer's suggestion, we have tried hard to establish Ewing sarcoma PDX models in our lab. We have obtained various Ewing sarcoma PDX tumors from Dr. Peter J. Houghton at UT Health San Antonio, including EW-5, NCH-EWS-1, UTH0584.000, UTH0589.000 commonly used in testing effects of various treatments on Ewing sarcoma tumor growth (Robles, et al. Clin Cancer Res. 2020. 26: 3012-3023). We thawed all tumors and isolated single cells for injection into nude mice to obtain fresh Ewing sarcoma PDX tumors. We were only able to recover the PDX tumor growth from NCH-EWS-1 PDX tumor cells. After obtaining fresh NCH-EWS-1 PDX tumors from nude mice, we further collected these tumors for single cell isolation. Afterwards, we injected single cell suspensions of NCH-EWS-1 PDX tumor cells with Matrigel into nude mice subcutaneously. When tumors are observable, we started treatment by vehicle, irinotecan (10 mg/kg, IP), MRX-2843 (10 mg/kg, IP) and the combination. Tumor growth was monitored by caliper measurements (revised Supplementary Fig. 10f). 27-days post-PDX cell injection, tumors were harvested and we found irinotecan significantly reduced NCH-EWS-1 PDX tumor growth, while the combination further reduced the PDX tumor growth (revised Supplementary Fig. 10f-10h) with minimal effects on animal weights (revised Supplementary Fig. 10i). These data suggest our proposed combination therapies demonstrate an efficacy in reducing Ewing sarcoma tumor growth in animal models derived from either Ewing sarcoma cell lines or PDX tumor cells.

14. Show that GAS6 is a direct target of STAT6? Or an indirect mechanism? Need to further prove or discuss.

Response: We thank the reviewer for raising this critical question. To test if STAT6 binds directly to GAS6 promoter, first of all, we bioinformatically analyzed if there are potential consensus STAT6 binding sites in 5' regulatory region of *GAS6* gene using Eukaryotic promoter database (SIB, Switzerland). 4 potential STAT6 binding motifs are predicted within -2kb region with a *p*-value smaller than 0.001, including SB1(-353), SB2(-487), SB3(-1440) and SB4(-1706). We then performed cut&run analyses using STAT6 antibody with MHH-ES-1 cells treated with DMSO or SN-38 for 24 hrs. STAT6 immunoprecipitated DNA fragments were analyzed by PCR. As indicated in the revised Fig. 5f, STAT6 demonstrated stronger binding capacity to SB1 and SB3 in MHH-ES-1 cells. Further SN-38 treatment significantly increased STAT6 occupancy on SB1 and SB3 (revised Fig. 5f). These data strongly suggest that STAT6 directly binds SB1 and SB3 in *GAS6* promoter, and SN-38 treatment enhances STAT6 localizing to SB1 and SB3, presumably through JAK1-mediated STAT6 phosphorylation.

15. Fig 3i: Check tumors treated with just irinotecan only for evidence of activation of these pathways/mechanisms of action..do you see evidence of the STAT6-GAS6-TAMK activation Compared to Fig 1m (where not clear of the tumor size or response to Irinotecan) the investigator should do analysis of tumors post-treatment---assess effects on small molecule targets?

Response: We thank the reviewer for bringing up this great question. Following the reviewer's suggestion, we have performed IHC analyses on isolated tumors treated with either irinotecan, UNC2025A or both. Consistent with revised Supplementary Fig. 2e, we observed increased ERK-p42/p44 and pSTAT6 signals in irinotecan treated tumors compared with vehicle control (revised Supplementary Fig. 5p). Moreover, UNC2025A treatment efficiently reduced both ERK-p42/p44 and STAT6-pY641 signals induced by irinotecan treatment (revised Supplementary Fig. 5p). We also tried *GAS6* staining in this IHC analysis, but we tried various *GAS6* antibodies and failed to find any signals. Since *Gas6* is a secreted

protein, we thought most secreted GAS6 proteins were lost during tumor fixation steps which yielded low signals. Nonetheless, these newly obtained IHC data support the notion that chemotherapy-induced JAK1/GAS6/TAM signaling may contribute to Ewing sarcoma tumor growth and UNC2025A reduces chemotherapy-induced JAK1/GAS6/TAM signaling to suppress Ewing sarcoma tumor growth.

16. Any evidence of these being present in relapsed patient tumors?

Response: We thank the reviewer for raising this excellent question. We would love to analyze if compared with primary tumors, relapsed Ewing sarcoma tumors display increased activation of the JAK/STAT6/GAS6/TAM signaling. However, given Ewing sarcoma is a rare pediatric cancer with ~200 cases each year in the US, and usually it is difficult to collect matched primary and relapsed tumors, and even it is difficult to distinguish between treatment naïve or resistant tumors, there is not enough clinical resources for this proposed direction. To date, there has been any noticeable databases for profiling protein changes associated with chemotherapy resistance in Ewing sarcoma.

Although various Ewing sarcoma cell lines at different stages of therapy were isolated including CHLA-9 at diagnosis, CHLA-10/25 after chemotherapy with progressive disease, and CHLA-258 post myeloablative therapy and autologous bone marrow transplantation, these lines from different disease/treatment stages didn't correlate with sensitivity to chemotherapies tested (May et al. PLOS ONE. 2013. 0080060), maybe because these cell lines were isolated from different patients with distinct genetic background. This suggests these established *in vitro* cell lines may not be faithful sources for studying chemoresistance. Thus, although this is an appealing and important question, due to the rareness of the disease and lack of clinical resources, we haven't been able to examine if any of our proposed signaling changes are also observed in relapsed, chemo-resistant Ewing sarcoma patients. We acknowledge this is a very important question to address and do hope more clinical resources will be available for Ewing sarcoma research in the future.

17. Fig 5: Seem like very high doses of SN-38 needed for effects on cell viability...see the effects of phosphorylation at low [], but not on viability when KO STAT6. What are the effects of phosphorylation at these higher doses of chemo? IC50s are 10fold higher than used for WBs. At these higher concentrations (1 0-300uM) what other signaling pathways are being altered? This needs to be discussed as well.

Response: We thank the reviewer for bringing up this question. We used relatively higher doses of SN-38 in cell viability assay in Fig. 5 because our cell viability assays were performed 48 hrs post-treatments. If we evaluate cell viability in a longer period (eg. a week) we can dramatically reduce the SN-38 doses to observe cell death. Following the reviewer's suggestion, we have performed western blotting analyses on the phosphorylation changes at these higher doses of chemo with a 24-hr treatment period. As shown in the revised Supplementary Fig. 7b, we observed that under these conditions, STAT6 depletion similarly reduced SN-38 induced Akt-pT308 and ERK-p42/p44 signals. In addition to the Akt/ERK signaling, we also examined Hippo signaling (Ahmed et al. J Cancer. 2015. 6(10): 1005-1010) and GSK3 β (Abe et al. J cancer Metastasis Treat. 2020. 6:51) phosphorylation that have been reported to be associated with chemotherapy treatment in Ewing sarcoma. To this end, we didn't find significant changes of these phosphorylation events either induced by SN-38 or regulated by STAT6 depletion (revised Supplementary Fig. 7b). These data suggest SN-38 or STAT6 is more likely to regulate specific protein phosphorylation events instead of globally changes on phospho-proteome. We acknowledge that more through phospho-profiling will answer this question, but it warrants a separate study. We have also included more statements in the discussion section for clarification.

18. Fig 8. In vivo studies would benefit from assessing the effects on more than 1 cell line model.

Response: We fully agree with the reviewer and following the reviewer's suggestion, we have also examined the efficacy of this combination therapy using Ewing sarcoma PDX murine models. PDX is a critical and more clinically related model to test the synergistic effects of our proposed combination therapy. Following the reviewer's suggestion, we have tried hard to establish Ewing sarcoma PDX models in our lab. We have obtained various Ewing sarcoma PDX tumors from Dr. Peter J. Houghton at UT Health San Antonio, including EW-5, NCH-EWS-1, UTH0584.000, UTH0589.000 commonly used in testing effects of various treatments on Ewing

sarcoma tumor growth (Robles, et al. Clin Cancer Res. 2020. 26: 3012-3023). We thawed all tumors and isolated single cells for injection into nude mice to obtain fresh Ewing sarcoma PDX tumors. We were only able to recover the PDX tumor growth from NCH-EWS-1 PDX tumor cells. After obtaining fresh NCH-EWS-1 PDX tumors from nude mice, we further collected these tumors for single cell isolation. Afterwards, we injected single cell suspensions of NCH-EWS-1 PDX tumor cells with Matrigel into nude mice subcutaneously. When tumors are observable, we started treatment by vehicle, irinotecan (10 mg/kg, IP), MRX-2843 (10 mg/kg, IP) and the combination. Tumor growth was monitored by caliper measurements (revised Supplementary Fig. 10f). 27-days post-PDX cell injection, tumors were harvested and we found irinotecan significantly reduced NCH-EWS-1 PDX tumor growth, while the combination further reduced the PDX tumor growth (revised Supplementary Fig. 10f-10h) with minimal effects on animal weights (revised Supplementary Fig. 10i). These data suggest our proposed combination therapies demonstrate an efficacy in reducing Ewing sarcoma tumor growth in animal models derived from either Ewing sarcoma cell lines or PDX tumor cells.

Reviewer #3

Yu and colleagues present an interesting study on signaling pathways that contribute to chemoresistance in Ewing sarcoma and provide data that nominate potential effective combination strategies. The authors provide evidence that chemotherapy induces phosphorylation/activation of multiple signaling proteins, notably AKT-T308, in Ewing sarcoma cells. The authors suggest that this chemo-induced signaling is dependent on TAM receptor activation via upregulation of Gas6 (presumably autocrine activation). The authors show that Gas6 induction is STAT6-mediated and that STAT6 is activated by JAK1, which itself is stimulated as a result of chemotherapy. Therapeutic implications are highlighted through xenograft studies where inhibition of JAK1 or TAM receptors in combination with chemotherapy potently reduces Ewing xenograft growth.

There are multiple noteworthy results. 1. Demonstration of a plausible pathway that induces TAM (Axl/MerTK) activation by chemotherapy; 2. Use of multiple Ewing cell lines increases rigor in the study; 3. Loss/Gain of function and pharmacologic inhibition of the TAM pathway results in consistent phenotypes in multiple cell lines, again increasing rigor; 3. Activation of JAK1 by chemo is a novel aspect of the study and the authors provide evidence to support this claim.

The work is original. There is limited published data on the contribution of Axl, MerTK, JAK1 and STAT6 individually in Ewing sarcoma. The concept that combination therapy is needed to improve outcomes in Ewing sarcoma is not novel. However, the strategy of looking for chemo-activated pathways in tumor cells and then targeting those pathways as a route to an effective combo is novel.

The methods are adequately described (except where noted below) and the study encompasses a wide breadth of techniques, results of which in general support the overall conclusions of the study.

Response: We thank the reviewer for acknowledging the importance and therapeutic potential for our study. We also thank the reviewer for raising these great suggestions to further improve our manuscript. Please find our detailed point-to-point responses to each raised question below.

Major Comments:

1. In general additional detail should be added to the figure legends and/or methods section where appropriate. For example, Figure 1B, the vendor and cat # for the membrane used to probe phospho-proteins should be provided and the conditions used to stimulate cells before lysate production should be clarified (e.g., how long were cells stimulated with SN-38). A key detail that should be provided in the figure legends is time frame of stimulation or assay.

Response: We thank the reviewer for raising this critical question and apologize for missing details for experiments. Following the reviewer's suggestions, in the revised manuscript, we have provided more details for experiments in the figure legend sections, and also have included a method section for the phospho-profiling assays.

2. Was AKT-pS473 included in the membrane shown in Fig 1B? The authors only show data with AKT-pT308, why? If AKT-s473 is not phosphorylated is the AKT pathway relevant to chemoresistance?

Response: We thank the reviewer for this great question. Akt-pS473 is also included in this array (two dots on the right of Akt-pT308 (labeled as D in Fig. 1b). There was slightly increase in Akt-pS473 upon SN-38 treatment as well, however, since the two replicates of Akt-pS473 dots were not as consistent as Akt-pT308, we thus relied on Akt-pT308 in the following analyses. In addition, although Akt-pS473 and Akt-pT308 are believed to be necessary for full Akt activation, it is thought Akt-pS473 primes for Akt-pT308 and the later accounts for more Akt activity (Manning et al. Cell. 2018. 169(3): 381-405). From our own experiences, in most cases Akt-pT308 and Akt-pS473 behave similarly in cells (Liu et al. Nature. 2014. 508 (9497): 541-545). Our previous study (Jiang et al. Nature Communications. 2019. 10: 1515) indicated activation of TAM kinases promotes both Akt-pS473

and pT308. Thus, in this study, we rely on Akt-pT308 as an Akt activation marker but we think Akt-pS473 behave similarly.

3. The activation of JAK1 by chemotherapy needs to be clarified. The text presents the concept that DNA damage response kinases (ATM, ATR, DNAPK) can phosphorylate JAK1 at S571 and this is repeated in the discussion. However, the authors do not mention that a SQ sequence is indicative of an ATM/ATR substrate site. It will be confusing for readers to just jump to JAK1-pSQ terminology. As written there are sections of the results that read as if JAK1 activation occurs in the absence of DNA damage response. Please clarify.

Response: We thank the reviewer for bringing up this critical question and fully agree with the reviewer that we should introduce SQ/TQ phosphorylation by DNA damaging kinases prior to presenting JAK1-pSQ data. In the revised manuscript, we have included background introductions for these DNA damage kinases and pSQ/pTQ phosphorylation motifs. In addition, we have also clarified that JAK1-pSQ requires DNA damage conditions in the revised manuscript.

4. Does activation of JAK1 by chemo occur in the nucleus or cytoplasm?

Response: We thank the reviewer for this great question. We performed new immunofluorescence microscope analyses on control or SN-38 treated MHH-ES-1 cells and found although SN-38 potentially promoted JAK1-pSQ in cells (revised Fig. 7d-7e), it didn't significantly affect JAK1 cellular localization (revised Supplementary Fig. 9h). In addition, we also included JAK1-S571A mutant as a negative control, and observed SN-38 also didn't regulate cellular localization of this phosphorylation-deficient mutant form of JAK1 revised Supplementary Fig. 9h). These data suggest that chemo-induced JAK1 activation didn't regulate JAK1 cellular localization.

To directly examine where chemo-induced JAK1-pSQ occurs, we performed cell fractionation assays and observed that SN-38-induced JAK1-pSQ largely occurred in the nucleus (revised Supplementary Fig. 9g).

5. The in vivo efficacy studies are promising. I strongly suggest/recommend repeating in vivo studies testing the best combo therapy vs chemo alone and control (limited groups) in mice with established tumors (>250 mm³) to drive higher impact.

Response: We thank the reviewer for this excellent suggestion and cannot agree more with the reviewer to further test effects of our proposed combination therapy on established large tumors to mimic clinical treatments. Following the reviewer's suggestion, we have xenografted MHH-ES-1 cells into nude mice and didn't start the treatments until tumors reached at least 250 mm³. As shown in the revised Supplementary Fig. 10b-10e, compared with each of the single treatment by irinotecan or MRX-2843, the combination of MRX-2843 with irinotecan significantly reduced tumor growth (revised Supplementary Fig. 10b-10d) with no observed toxicity to animals (revised Supplementary Fig. 10e). We fully agree with the reviewer that these newly included data further strengthened our confidence in applying proposed combination therapy in treating Ewing sarcoma in clinical trials.

Minor:

6. In Fig 2 the authors show that KD of AXL blocks chemo-induced AKT-pT308 but cabozantinib, a reasonably potent inhibitor of AXL, does not. This is somewhat confusing and should be discussed.

Response: We thank the reviewer for bringing up this question. We apologize for the typo in the original manuscript as AXL depletion didn't block SN-38 induced Akt-pT308 in MHH-ES-1 cells (revised Fig. 2e), nor increased Akt-pT308 by additional chemotherapeutics including doxorubicin (revised Supplementary Fig. 3b), etoposide (revised Supplementary Fig. 3e) nor TMZ (revised Supplementary Fig. 3h). We have corrected this statement in the revised manuscript.

7. I recommend showing a cell viability curve for at least a few of the combination cell viability studies displayed in Fig 3a-h. Just as examples, these could be in the supplement. The heat maps are not intuitive.

Response: We thank the reviewer for raising this question and following the reviewer's suggestion, we have included representative cell viability data for the SN-38+irinotecan combination from MHH-ES-1, A673 and TC-71 cells in the revised Supplementary Fig. 5a. We have also included more details for this experiment in the corresponding figure legends.

8. I would avoid the use of JAK1-KD terminology. 'KD' is commonly used for knockdown or kinase dead, which can easily lead to reader confusion.

Response: We thank the reviewer for raising this great suggestion and apologize for the confusions. We have replaced KD (kinase domain) with KM (kinase motif) to alleviate this concern in the revised manuscript.

9. Fig 6c is unconvincing as displayed. Either replace, move to supplement or delete. The cell fractionation is sufficient to demonstrate nuclear localization.

Response: We thank the reviewer for bringing up this critical concern and apologize for the low-quality data in Fig. 6c. We have repeated this assay and replaced original figure panels with newly obtained images with quantifications of nuclear/cytosolic localized signals in the revised Fig. 6c.

Suggested edits:

Line 237 – change 'BCA' to 'TCA'

Response: We have replaced "BCA" with "TCA" and apologize for this typo.

Line 260 – change 'macrophage' to 'macrophages'

Response: We have replaced "macrophage" with "macrophages".

Reviewer #4

This manuscript reports very interesting studies on the understanding of the molecular basis of Ewing sarcoma chemoresistance. The authors provide strong data to suggest that chemotherapy induces JAK1 activation via DNA damaging kinases-mediated JAK1-SQ phosphorylation, which leads to the subsequent activation of STAT6, resulting in the transcriptional activation of GAS6. The secretion of GAS6 protein induces TAM kinase activation and then the increased activation of Akt and ERK in Ewing sarcoma cells. The authors propose that this newly identified JAK1/STAT6/GAS6/TAM/Akt-ERK pathway contributes to Ewing sarcoma chemoresistance. In support of their hypothesis, JAK1 inhibitor filgotinib or TAM kinase inhibitor UNC2025A sensitized Ewing sarcoma cells and tumors to chemotherapy *in vitro* as well as in animal tumor models. In addition, the TAM kinase inhibitor MRX-2843 currently in clinical trials to treat AML and advanced solid tumors showed synergistic effects with chemotherapy to inhibit Ewing sarcoma tumor growth. These results have potential translational value to overcome Ewing sarcoma chemoresistance. There are a few issues that the authors should address.

Response: We thank the reviewer for acknowledging the importance and therapeutic potential for our study. We also thank the reviewer for raising these great suggestions to further improve our manuscript. Please find our detailed point-to-point responses to each raised question below.

1) Several previous studies have reported the involvement of STAT3 in the development of chemoresistance, and GAS6 is known to induce the activation of STAT3. The authors should provide experimental data to address if STAT3 plays any role in Ewing sarcoma chemoresistance.

Response: We thank the reviewer for raising this excellent question and fully agree that previous reports showing GAS6 induces STAT3 activity to facilitate mesangial cell proliferation (Yanagita et al. J Biol Chem. 2001. 276(45):42364-42369), and STAT3 activation contributes to chemoresistance in multiple types of cancers (Yang et al. Cancers. 2020. 12(9): 2459). To examine if STAT3 plays a role in Ewing sarcoma chemo-resistance, we generated STAT3-depleted MHH-ES-1 cells and found STAT3 depletion didn't affect SN-38-induced Akt-pT308 and ERK-p42/p44 (revised Supplementary Fig. 8h-8i). Moreover, STAT3 depletion also didn't significantly affect MHH-ES-1 cell responses to SN-38 treatment *in vitro* (revised Supplementary Fig. 8j). These data suggest that in Ewing sarcoma, STAT3 is less likely to play a role in regulating chemotherapy responses.

2) Have the authors tested if JAK1 becomes tyrosine phosphorylated/activated by chemo drugs in Ewing sarcoma cells?

Response: We thank the reviewer for raising this excellent question. We have examined JAK1-pY1034/pY1035 in MHH-ES-1 cells treated with SN-38 or etoposide, and observed that neither of these chemotherapeutic agents significantly increased JAK1-pY signals (revised Supplementary Fig. 9a-9c maybe mild increases). These newly obtained data suggest chemotherapy-induced JAK1 activation is different from IL-mediated JAK1 activation through FERM/SH2 domain mediated JAK1-pY signals, rather more likely through JAK1-pS571Q to induce JAK1 activation.

3) Can the authors show the direct binding of STAT6 to the promoter of GAS6 gene by ChIP assays?

Response: We thank the reviewer for raising this critical question. To test if STAT6 binds directly to GAS6 promoter, first of all, we bioinformatically analyzed if there are potential consensus STAT6 binding sites in 5' regulatory region of Gas6 gene using Eukaryotic promoter database (SIB, Switzerland). 4 potential STAT6 binding motifs are predicted within -2kb region with a *p*-value smaller than 0.001, including SB1(-353), SB2(-487), SB3(-1440) and SB4(-1706). We then performed cut&run analyses using STAT6 antibody using MHH-ES-1 cells treated with DMSO or SN-38 for 24 hrs. STAT6 immunoprecipitated DNA fragments were analyzed by synthesized PCR primers specifically for SB1, SB2, SB3 or SB4. As indicated in the revised Fig. 5f, STAT6 demonstrated stronger binding capacity to SB1 and SB3 in MHH-ES-1 cells. Further SN-38 treatment significantly increased STAT6 occupancy on SB1 and SB3 (revised Fig. 5f). These data strongly suggest that STAT6 might directly bind SB1 and SB3 in GAS6 promoter, and SN-38 treatment enhances STAT6 localizing

to SB1 and SB3, presumably through JAK1-mediated STAT6 phosphorylation.

4) Serine phosphorylation of STATs is important for STAT-mediated gene activation. Several kinases including ERKs are involved in STAT serine phosphorylation. Since chemotherapy was shown to induce ERK activation in Ewing sarcoma cells, does the chemotherapy-induced ERK activation have any effect on STAT serine phosphorylation, if so, does this phenomenon contribute to the chemoresistance in Ewing sarcoma?

Response: We thank the reviewer for raising this critical question. We fully agree with the reviewer that in addition to STAT6-pY641, STAT6 can also be phosphorylated by additional kinases such as TBK1 (Chen et al. Cell. 2011. 147(2): 436-446), ERK (So et al. Mol Immunol. 2007. 44(13): 3416-3426) or JNK (Shirakawa et al. J Biol Chem. 2010. 286(5): 4003-4010) on serine residues that differentially regulates its transcriptional activity. Specifically, TBK1 phosphorylates STAT6 at Ser407 to facilitate its Y641 phosphorylation and activation. ERK is critical for IL-induced STAT6 activation. JNK phosphorylates STAT6-Ser707 to suppress STAT6 binding with DNA thus inactivating STAT6. Due to the lack of specific phospho-serine antibodies for ERK-mediated STAT6 phosphorylation, we relied on the pSer-PKC substrate antibody (Cell Signaling Technology #2261) for this test. As shown in the Fig. R2 below, we didn't observe any changes on STAT6-pSer signals. In addition, we found that ERK inhibition by PD0325901 failed to reduce SN-38-induced GAS6 transcription in MHH-ES-1 cells (revised Supplementary Fig. 8f), suggesting ERK and ERK-mediated STAT6 Ser phosphorylation may not play a critical role here. Consistently, PD0325901 didn't affect SN-38 induced STAT6-pY641 signals in MHH-ES-1 cells (revised Supplementary Fig. 8g). These data suggest ERK/STAT6 signaling may not contribute to Ewing sarcoma chemoresistance.

Figure R2. IB analyses of HA-IPs and WCL from MHH-ES-1 cells stably expressing HA-STAT6 by lenti-viral infection.

Reviewers' Comments:

Reviewer #1:

Remarks to the Author:

The authors have made good attempts at addressing most of my comments, including through repeat experimentation as well as text correction – I appreciate this. I have a few further comment on issues which I believe remain to be addressed and clarified, in order to make the manuscript finally acceptable for publication.

Furthermore, in any further revised version of the manuscript, I urge the authors to assist the reviewers by stating precisely where in the revised manuscript the stated changes have been made (e.g. Page, line numbers etc.).

1. The authors have now stated info on the nos. of repeat W blot expts. However, I am afraid that I cannot see anywhere in the revised manuscript where quantifications of W blots from densitometry have been performed (e.g. in revised Figs. 2, 4, Suppl Fig. 3).

2. In answer to my comment 2 on the cell lines, the authors responded "Cell lines were verified by the manufacturer's websites and regularly checked by morphology for authentication". The level of authentication that I was referring to, and which is most often that expected by major journals, is by STR genotyping analysis. Microscopic investigation is I am afraid limited in its reliability for cell line verification.

3. In answer to my comment 5, The new, repeat W blot expts now do show that MERTK depletion reduces pERK signals. However, I still do not see from the W blot in Fig. 2E that the AXL depletion has any effect on pERK. Therefore, it is wise for the authors to discuss the fact that the growth inhibition they see from AXL knockdown may well not be due to ERK signalling.

Again, I see no quantitative data derived from W blot band quantification, as the authors state.

4. In answer to my comments 6 and 7, the authors have used a more appropriate wording of the effects observed; however, again, I see no quantitative data derived from W blot band quantification, as the authors state.

5. In answer to my comment 16 on nuclear translocation, the repeat microscopic analysis is now visually clearer and more convincing, alongside quantification of the nuclear:cytoplasmic ratios. However, the method and quantitation needs to be described, and more detail provided in the Fig. 6 legend.

Also, on page 13, the authors write "Independent IF experiments also confirmed that SN-38 treatment induced STAT6 nuclear translocation (Fig. 6c and Supplementary Fig. 8a). There is no such relation to what is shown in Supplementary Fig. 8a.

Reviewer #2:

Remarks to the Author:

Overall Yu and colleagues have attempted to address many of the comments and critiques, including the use of testing their mechanisms on more cell line models. Again, they provide very good mechanistic data demonstrating that p-ERK and p-AKT are potential adaptive mechanisms of resistance to chemotherapy. A couple of issues still remain, including that it does not appear the degree of p-AKT/p-ERK is associated with resistance to SN-38 (as shown in rebuttal letter)--SN1 with low p-AKT and p-ERK still as "resistant" as SN2 and 3 that have higher p-levels. It would be appropriate to further discuss these results.

Also, the isoflavone data does not add much and actually brings in a little confusion.

1. It is not really used (this reviewer has not ever given a patient isoflavone as upfront therapy) and 2. Interesting that as a hormone/estrogen analogue it would have the same mech of resist as the DNA targeting chemos (Fig 8, panel p)? Thoughts/comments on this?

Final overall concern is the lack of evidence for these mechanisms of resistance/decreased sensitivity being present in refractory/relapsed EWS patient tumors.

IS there any transcriptomic data the authors can use from pre- and post patients to determine if these pathways are activated (STAT6 targets, increased Gas6 expression etc...). Honestly not sure

if that data is out there, but might be.

Reviewer #3:

Remarks to the Author:

The authors have done a commendable job addressing the prior concerns. I have not further queries.

Reviewer #4:

Remarks to the Author:

The authors have now fully addressed my previous concerns on this revised manuscript.

Point-to-Point Responses to the Reviewers' Critiques (NCOMMS-23-43600A)

We deeply appreciate the constructive suggestions provided by the editor and four reviewers during the second review of our manuscript. We are glad to find reviewers are largely satisfied with our revised data. Over the past 1/2 month, we have obtained additional experimental evidence to address the remaining concerns from two reviewers and to further support our conclusions. We hope that the editor and the reviewers concur with us that we have fully addressed all the reviewers' concerns and substantially strengthened our paper. Therefore, we believe that the revised manuscript is now suitable for publication in *Nature Communications*.

Reviewer #1 (Remarks to the Author):

The authors have made good attempts at addressing most of my comments, including through repeat experimentation as well as text correction – I appreciate this. I have a few further comment on issues which I believe remain to be addressed and clarified, in order to make the manuscript finally acceptable for publication. Furthermore, in any further revised version of the manuscript, I urge the authors to assist the reviewers by stating precisely where in the revised manuscript the stated changes have been made (e.g. Page, line numbers etc.).

Response: We thank the reviewer for acknowledging our efforts in addressing previous raised concerns. We also thank the reviewer for further instructive suggestions. As instructed, we included page and line number for changes we made during this round of revision in the revised manuscript.

1. The authors have now stated info on the nos. of repeat W blot expts. However, I am afraid that I cannot see anywhere in the revised manuscript where quantifications of W blots from densitometry have been performed (e.g. in revised Figs. 2, 4, Suppl Fig. 3).

Response: We thank the reviewer raising this important suggestion. We have provided quantifications for western blot analyses in the revised Fig. 2d-2f, Fig. 4n, 4o and revised Supplementary Fig. 3a-3l. Please note the quantifications for densitometry are for Akt-pT308 signals normalized to total Akt1 proteins and pERK signals normalized to total ERK proteins.

2. In answer to my comment 2 on the cell lines, the authors responded “Cell lines were verified by the manufacturer's websites and regularly checked by morphology for authentication”. The level of authentication that I was referring to, and which is most often that expected by major journals, is by STR genotyping analysis. Microscopic investigation is I am afraid limited in its reliability for cell line verification.

Response: We thank the reviewer for raising this question. A673, MHH-ES-1 and SK-N-MC cells have been recently authenticated by the Davis lab by STR approaches. We have included these information in the method section (page 25 line 568). Given most of studies in this manuscript focus on MHH-ES-1 cells, we are certain MHH-ES-1 cells are correctly authenticated.

3. In answer to my comment 5, The new, repeat W blot expts now do show that MERTK depletion reduces pERK signals. However, I still do not see from the W blot in Fig. 2E that the AXL depletion has any effect on pERK. Therefore, it is wise for the authors to discuss the fact that the growth inhibition they see from AXL knockdown may well not be due to ERK signalling. Again, I see no quantitative data derived from W blot band quantification, as the authors state.

Response: We thank the reviewer for bringing up this question. We agree with the reviewer that ALX depletion in Ewing sarcoma lines we tested didn't significantly reduce ERK activity at basal levels (revised Fig. 2e), thus, reduced growth of AXL depleted MHH-ES-1 cells (revised Supplemental Fig. 4b) could not be explained by decreased ERK activation. Following the reviewer's suggestion, we have included more sentences to clarify this

point in the discussion to clarify this point (Page 8, line 187 in the revised manuscript). We have provided quantifications for western blot analyses in the revised Fig. 2d-2f, Fig. 4n, 4o and revised Supplementary Fig. 3a-3l. Please note the quantifications for densitometry are for Akt-pT308 signals normalized to total Akt1 proteins and pERK signals normalized to total ERK proteins.

4. In answer to my comments 6 and 7, the authors have used a more appropriate wording of the effects observed; however, again, I see no quantitative data derived from W blot band quantification, as the authors state.

Response: We thank the reviewer's suggestion. We have included quantifications of densitometry for ERK-p42/p44 signals normalized to total ERK as in the revised Fig. 4n and 4o.

5. In answer to my comment 16 on nuclear translocation, the repeat microscopic analysis is now visually clearer and more convincing, alongside quantification of the nuclear:cytoplasmic ratios. However, the method and quantitation needs to be described, and more detail provided in the Fig. 6 legend.

Also, on page 13, the authors write "Independent IF experiments also confirmed that SN-38 treatment induced STAT6 nuclear translocation (Fig. 6c and Supplementary Fig. 8a). There is no such relation to what is shown in Supplementary Fig. 8a.

Response: We thank the reviewers for acknowledging our efforts to illustrate STAT6 nuclear translation upon chemotherapy. Following the reviewer's instruction, we have included the quantification method in the Fig. 6 legend on Page 40, line 1182 in the revised manuscript.

The results in Supplementary Fig. 8a are a confirmation for the MHH-ES-1 cell line stably expressing HA-STAT6 used for the IF experiment in Fig. 6c. We always validate the cells prior to experiments. To clarify this experiment and data panel, we have revised the main text (page 13 line 306-308) to clearly indicate the results.

Reviewer #2 (Remarks to the Author):

Overall Yu and colleagues have attempted to address many of the comments and critiques, including the use of testing their mechanisms on more cell line models. Again, they provide very good mechanistic data demonstrating that p-ERK and p-AKT are potential adaptive mechanisms of resistance to chemotherapy.

1. A couple of issues still remain, including that it does not appear the degree of p-AKT/p-ERK is associated with resistance to SN-38 (as shown in rebuttal letter)--SN1 with low p-AKT and p-ERK still as "resistant" as SN2 and 3 that have higher p-levels. It would be appropriate to further discuss these results.

Response: We thank the reviewer for raising this question. We agree with the reviewer that in the SN-38 resistant MHH-ES-1 single clones, the levels of increased pAkt/pERK do not correlate well with their resistance ability. We are of the opinion that changes of pAKT/pERK signaling may be only one outcome within many other changes associated with chemoresistance in these single clones, as heterogeneity in chemo-resistant clones are frequently observed (Zhang *et al.* 2022. *Int J Biol Sci.* 18(7): 3019-3033). Whether these increased pAKT/pERK signaling contributes to chemoresistance in these clones remain unclear. These data are not integrated into this study largely because these *in vitro* cell line work could not convincingly demonstrate if activated AKT/ERK signaling contributes to chemoresistance in clinic.

Interestingly, a recent study determined the transcriptome changes in 4 Ewing sarcoma patients with good responses to chemotherapy vs. 9 patients with poor responses (Chen *et al.* 2020. *Mol Oncol.* 14(5): 1101-1117). Re-analyzing this published data led us to find that increased JAK1 and STAT6 expression were identified in these patients positively correlated with poorer response to chemotherapy (revised Supplementary Fig. 10j). However, due to the small number of patients in this study, we cannot conclude if increased JAK1/STAT6 expression contributes to Ewing sarcoma chemoresistance.

We are of the opinion that our proposed chemotherapy/JAK1/STAT6/GAS6/TAM/Akt/ERK signaling may not directly contribute to chemoresistance as a stable deregulated signaling, rather activated Akt/ERK signaling will provide a protection for Ewing sarcoma cells to develop genetic changes to confer stable/acquired resistance to chemotherapy. Otherwise, cells will be killed by chemotherapy even before acquired resistance can be developed. We have also included these statements in the discussion section (page 23 line 532).

2. Also, the isoflavone data does not add much and actually brings in a little confusion. It is not really used (this reviewer has not ever given a patient isoflavone as upfront therapy) and 2. Interesting that as a hormone/estrogen analogue it would have the same mech of resist as the DNA targeting chemos (Fig 8, panel p)? Thoughts/comments on this?

Response: We thank the reviewer for raising this question. Although infrequent, isoflavone has also been used as an alternative in upfront therapy (Zollner, S.K., *et al.* *J Clin Med.* 2021. 10). Previous study demonstrated that estrogen- and stress-induced DNA damage is an important factor in the development and progression of breast cancer, and isoflavone supplementation reduces oxidative DNA damage in healthy women (Daniela Erba *et al.* *Nutrition Research.* 2012. 32(4):233-240). Therefore, we speculate that isoflavones may function similarly induce DNA damage, thereby activating the JAK1/STAT6/GAS6/TAM signaling in Ewing sarcoma. Since these data are not essential to our conclusions, to avoid future confusions, we have removed the isoflavone data from the revised manuscript (original supplemental Figs. S11-S1q on page 6 line 135).

3. Final overall concern is the lack of evidence for these mechanisms of resistance/decreased sensitivity being present in refractory/relapsed EWS patient tumors. IS there any transcriptomic data the authors can use from pre- and post patients to determine if these pathways are activated (STAT6 targets, increased Gas6 expression etc...). Honestly not sure if that data is out there, but might be.

Response: We thank the reviewer for bringing up this excellent question, and fully agree that it is critical to confirm if our proposed signaling deregulation could be observed in relapsed patients. However, given that Ewing sarcoma is a rare pediatric disease with only ~200 patients in the US annually, there hasn't been any cohort of

patient samples for paired prior- and post-chemotherapy established nation-wide. In addition, given when relapse occurs the recurrent tumors may not be at the same location, whether changes in the recurrent tumors contribute to chemoresistance remains unclear.

To date, there have been extensive efforts using microarray or RNA-Seq approaches to define prognostic markers for Ewing sarcoma, including NCBI Gene Expression Omnibus (GEO) databases as listed below: GSE63155 (46 Ewing sarcoma patients, Volchenboum et al. 2015. *J Pathol Clin Res.* 1(2): 83-94), GSE63166 (39 Ewing sarcoma patients, Modzelewski et al, 2015, *J Cell Sci.* 128(12): 2314-2127), GSE17679 (32 Ewing sarcoma patients, Savola et al. 2011. *ISRN Oncol.* 168712), and GSE34620 (38 Ewing sarcoma patients, Postel-Vinay et al. 2012. *Nat Genet.* 44(3): 323-327), GSE12102 (30 primary Ewing sarcoma tumors and 7 metastatic tumors, Scotland, et al. 2009. *J Clin Oncol.* 27(13): 2209-2216) and others. Based on these analyses, multiple prognostic markers have been proposed but their contributions to chemoresistance are unknown.

Due to the rareness of the disease, prior work in finding Ewing sarcoma chemotherapy resistance mechanisms was initiated from pan-cancer levels or from Ewing sarcoma cell lines. These efforts include data mining of chemoresistance genes from all cancer types followed by validation of Ewing sarcoma related pathways by developing chemo-resistant Ewing sarcoma cell line models (Horbach, et al. 2018. *Mol Clin Oncol.* 8(6): 719-724); using RNA-Seq to find differential gene expression in CD133 high or low Ewing sarcoma cell lines that establish a correlation of stemness with chemoresistance (Roundhill et al. 2023. *Cancers.* 15(3): 769) and using Ewing sarcoma PDX models (by establishing three pairs of Ewing sarcoma PDX models from the same patients at primary diagnosis and later stages and focused on CNA (copy number alternations) and SN-38 distributions with no intention to look into chemoresistance profiles (Castillo-Ecija et al. 2020. *Journal of Controlled Release.* 324 (10): 440-449)). These studies provide further insights into possible chemo-resistant mechanisms; however, clinical validations would be necessary.

Interestingly, a recent study determined the transcriptome changes in 4 Ewing sarcoma patients with good responses to chemotherapy vs. 9 patients with poor responses (Chen *et al.* 2020. *Mol Oncol.* 14(5): 1101-1117). Re-analyzing this published data led us to find that increased JAK1 and STAT6 expression were identified in these patients positively correlated with poorer response to chemotherapy (revised Supplementary Fig. 10j). However, due to the small number of patients in this study, we cannot conclude if increased JAK1/STAT6 expression contributes to Ewing sarcoma chemoresistance.

In addition, we are of the opinion that our proposed chemotherapy/JAK1/STAT6/GAS6/TAM/Akt/ERK signaling may not directly contribute to chemoresistance as a stable deregulated signaling, rather activated Akt/ERK signaling will provide a protection for Ewing sarcoma cells to develop genetic changes to confer stable/acquired resistance to chemotherapy. Otherwise, cells will be killed by chemotherapy even before acquired resistance can be developed. We have also included these statements in the discussion section (page 23 line 532).

We hope the reviewer agrees with us that due to the lack of clinical resources for this rare disease, and our proposed signaling may not directly play roles in either intrinsic or acquired chemoresistance in Ewing sarcoma, if deregulation of proposed chemotherapy/JAK1/STAT6/GAS6/TAM/AKT/ERK signaling is observed in relapsed or chemo-resistant Ewing sarcoma patients warrants further investigations in a separate study.

Reviewer #3 (Remarks to the Author):

The authors have done a commendable job addressing the prior concerns. I have not further queries.

Response: We thank the reviewer for insightful suggestions to improve our study and acknowledging our efforts to address raised concerns.

Reviewer #4 (Remarks to the Author):

The authors have now fully addressed my previous concerns on this revised manuscript.

Response: We thank the reviewer for constructive suggestions to further improve our study and acknowledging our efforts to address raised concerns.

Reviewers' Comments:

Reviewer #1:

Remarks to the Author:

The authors have now performed the analysis that was missing in the first revision. They have also made clarifications/descriptions in the revision where appropriate, as directed.

The manuscript is in my view now suitable for publication. I would only remark now that, in the proofs, the authors can briefly add the method by which they did the Western blot band densitometry, as this is missing from the Methods section/Fig legends.

Congratulations to the authors for this fine work.

Reviewer #2:

Remarks to the Author:

The authors have addressed the critiques.

Point-to-Point Responses to the Reviewers' Critiques (NCOMMS-23-43600B)

We deeply appreciate the constructive suggestions provided by the editor and two reviewers during the third review of our manuscript. We are glad to find both reviewers are satisfied with our revised data. In the past week, we have included the western blot quantification method information in the revised manuscript as suggested. We hope that the editor and the reviewers concur with us that we have fully addressed all the reviewers' concerns and substantially strengthened our paper. Therefore, we believe that the revised manuscript is now suitable for publication in *Nature Communications*.

Reviewer #1 (Remarks to the Author):

The authors have now performed the analysis that was missing in the first revision. They have also made clarifications/descriptions in the revision where appropriate, as directed. The manuscript is in my view now suitable for publication. I would only remark now that, in the proofs, the authors can briefly add the method by which they did the Western blot band densitometry, as this is missing from the Methods section/Fig legends. Congratulations to the authors for this fine work.

Response: We thank the reviewer for constructive suggestions in previous reviews of our manuscript. Following the reviewer's suggestion, we have included the method and details for western blot band densitometry determination in the revised method section.

Reviewer #2 (Remarks to the Author):

The authors have addressed the critiques.

Response: We thank the reviewer for helping us further improve our manuscript in the review process.